# FeedSign: Full-parameter Federated Fine-tuning of Large Models with Extremely Low Communication Overhead of One Bit

## Abstract

Federated fine-tuning (FFT) aims to fine-tune a pre-trained model with private data from distributed clients by exchanging models rather than data under the orchestration of a parameter server (PS). However, as large models are acing in almost every machine learning task, the communication overhead and memory demand are surging accordingly, hindering the practical deployment on consumer devices. To overcome the bottleneck forged by the growing communication overhead of federated learning and lower the high memory demand of large model fine-tuning, we propose FeedSign, an FFT algorithm where a client uploads its update model and downloads the global model of any size using exactly 1 bit per step, while the memory demand is squeezed to the amount needed for inference. This is realized by utilizing zeroth-order (ZO) optimizers on large models and shared pseudo-random number generators (PRNG) across devices to split the gradient estimate from the clients to 1) a direction corresponding to a designated random seed and 2) a binary vote from the client indicating whether the seed-corresponding direction grants a local loss descent, which is the only information the clients should convey to the PS. We conduct theoretical analysis on FeedSign and show that it converges at an exponential rate $\mathcal{O}(e^{-t})$, where $t$ is the number of elapsed steps, the same rate as in first-order (FO) methods can attain in big $\mathcal{O}$ notation. Moreover, it is also found that FeedSign enjoys good robustness against data heterogeneity and Byzantine attacks. We conduct extensive experiments on models across different structures and sizes (11M to 13B) and found that the proposed method performs better or closely, depending on scenarios, compared to its ZO and FO counterparts albeit an orders-of-magnitude lower communication overhead. We also discuss some interesting advantages as byproducts guaranteed by the minimalistic design of FeedSign.

## 1 Introduction

The development of deep learning (DL) has allowed us to enjoy better intelligent services by training larger models on broader data. While large models demonstrate good performance in general cases, they hold promise to provide more tailored services and greatly improve their ability on intelligent applications if the rich but privacy-sensitive local data of users can be accessed for learning. Federated learning (FL) McMahan et al. (2017) stands out as a solution to achieve privacy-preserving distributed learning by frequently averaging the model parameters generated by the local data but leaving the data intact at their holders. The algorithm is known as federated averaging (*FedAvg*). When the paradigm is applied on a fine-tuning task, it is often termed federated fine-tuning (FFT) Popov et al. (2018).

Such a learning paradigm demands that stochastic gradient descent (SGD) algorithms to be run on client devices. However, the assumed participating client devices are usually resource-restrictive devices like phones and tablets, where SGD can be a heavy computation burden. Moreover, as large models become increasingly popular due to their versatility and great performance, model update aggregation under the FFT paradigm becomes prohibitively expensive. Different methods have been proposed to lower communication and computation costs. Pioneering works include model splitting Thapa et al. (2022) that proposed to split the DL model into two parts so that most of the computation

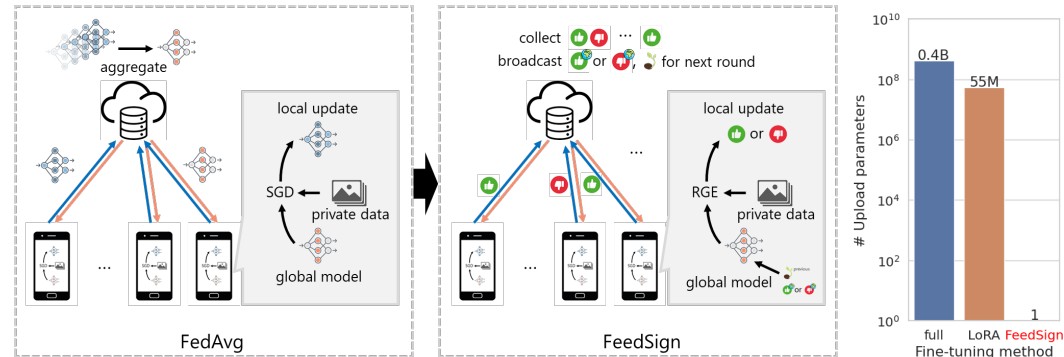

Figure 1: *Left*: Overview of *FedAvg* and *FeedSign*; *Right*: Comparison of *FedAvg* and *FeedSign* in terms of communication cost, measured by the number of parameters communicated in a communication round, taking FFT task on a RoBERTa-large model as an example. The $y$-axis is in **logarithmic** scale.

overhead can be unloaded to the PS as well as reducing communication costs. As for the special case of large models, one of the most successful methods is Parameter-efficient Fine-tuning (PEFT), focusing on updating only a small part of the large models hence lowering the communication and computation costs. Some of the techniques include LoRA Hu et al. (2021), Prefix Fine-tuning Li & Liang (2021), BitFit Zaken et al. (2021) and Adapter Narayanan et al. (2021); Pfeiffer et al. (2020).

However, while the aforementioned methods hold the promise of largely reducing the number of trainable parameters (which scales to the communication cost) with little performance drop, the communication cost of FFT tasks is still formidable. As a qualitative comparison, to participate in FFT on a RoBERTa-large model, a client device will upload around 53 million float numbers during a communication round, which takes up around 100 MB, the size of 5 minutes of YouTube full high definition (FHD, 1080p) video, whereas FFT usually takes thousands of communication rounds to converge, apart from the huge memory demand.

A series of pioneering works Xu et al. (2024); Qin et al. (2023) leverages zeroth-order optimization and the shared Pseudo Random Number Generators (PRNG) across modern deep learning frameworks like PyTorch Paszke et al. (2019) and Tensorflow Abadi et al. (2016) to lower the per-step uplink communication overhead to KB level and the memory demand to an almost equal amount of that of inference. However, we show that the per-step uplink and downlink communication overhead can be further reduced to 1 bit per step regardless of model size with little performance loss but several advantages, including data heterogeneity, Byzantine resilience, and parameter security. Specifically, our contributions are as follows:

1. Establishing upon MeZO Malladi et al. (2023), we proposed *FeedSign*, an FFT framework compatible with both full-parameter fine-tuning and PEFT, featuring per-step uplink communication overhead of **1 bit** and inference-level memory demand, regardless of model size. This is realized by utilizing zeroth-order (ZO) optimizers on large models and shared pseudo-random number generators (PRNG) across devices to split the gradient estimate from the clients to 1) a direction corresponding to a designated random seed and 2) a binary vote from the client indicating whether the seed-corresponding direction grants a local loss descent, which is the only information the clients should convey to the PS.

2. We provide the convergence analysis of our method. We found that it converges at an exponential rate $\mathcal{O}(e^{-t})$, the same rate as in first-order (FO) methods can attain in big $\mathcal{O}$ notation, where $t$ is the number of elapsed steps. The analysis implies that *FeedSign* has surprising effects addressing some long-standing problems of FL, including communication bottleneck, data heterogeneity, and Byzantine vulnerability.

3. We conduct comprehensive experiments across different model types (ResNet, ViT, RoBERTa, and OPT) and scales (11M to 13B) to verify the performance of *FeedSign* across various downstream language and vision tasks. It is observed that,

   (a) Compared with the conventional FO counterpart, with **close-to-zero** communication overhead regardless of the model size (1 bit versus 24 GB per step for OPT-13B)

and inference-level memory (around $1/12$ for transformer-based models Malladi et al. (2023)), *FeedSign* achieves comparable test performance;

(b) Compared with federated ZO baselines, with at most $1/64$ of communication overhead, *FeedSign* achieves comparable test performance in general settings while outperforming remarkably under data heterogeneity and Byzantine attacks.

4. We discuss some interesting features as byproducts that *FeedSign* will bring to a FL system on parameter security, hardware requirements, and differential privacy.

## 2 RELATED WORKS

### 2.1 FEDERATED LEARNING

Federated learning (FL) contrasts with centralized learning by training a shared model using data from distributed owners without directly sharing the data thereby preserving data privacy McMahan et al. (2017). Although centralized learning usually provides an upper bound of performance, FL has its unique advantages as it can access data originally unavailable to centralized learning due to privacy concerns Yang et al. (2018); Hard et al. (2018); Cormode et al. (2018). It is also suitable for uniting siloed raw data without compromising confidentiality as in various fields like healthcare Ogier du Terrail et al. (2022); Rieke et al. (2020) and financing Long et al. (2020).

However, FL's privacy protection comes at the cost of frequent model parameter exchanges, creating a communication bottleneck Konečný et al. (2016); Kairouz et al. (2021). The success of large pre-trained models in various tasks Liu et al. (2019); Achiam et al. (2023); Jiang et al. (2023a); Dosovitskiy et al. (2020) highlights the need to address this bottleneck. Parameter-efficient fine-tuning techniques, which can reduce the number of trainable parameters, show promise when combined with FL to minimize communication overhead Sun et al. (2024); Cho et al. (2023); Zhang et al. (2023); Kim et al. (2023). However, the communication overhead inevitably scales to the number of trainable parameters in all of methods above.

### 2.2 ZO OPTIMIZATION FOR DL AND FL

Over the years, FO methods like SGD and its variants have been the default choice for DL model training Gardner (1984); Amari (1993); Bottou (2010); Kingma (2014); Bottou et al. (2018). This method aims to minimize an objective $\mathcal{L}(\boldsymbol{w})$ that characterizes how bad a function $f_{\boldsymbol{w}}$ parameterized by a numerical vector $\boldsymbol{w}$ is mapping from an input space $\mathcal{X}$ to an output space $\mathcal{Y}$ using the chain rule and automatic differentiation Griewank (2014); Paszke et al. (2017) to approach the derivative of $\mathcal{L}(\boldsymbol{w})$ with respect to $\boldsymbol{w}$. Nonetheless, some objectives of interest are non-differentiable or whose gradients are expensive to compute calling for alternatives. They are usually known as ZO optimization since they do not require explicit gradient information for objective minimization.

The combination of ZO optimization and FL has been a hot research topic in recent years since in FL settings clients are usually resource-limited and ZO can make the estimation of gradients less expensive Fang et al. (2022); Qiu et al. (2023); Chen et al. (2024a); Ling et al. (2024); Maritan et al. (2024). However, the communication bottleneck is still a huge problem for real deployment.

Notably, *FwdLLM* Xu et al. (2024) and *FedKSeed* Qin et al. (2023) are the closest works to ours, where the authors discuss a federated fine-tuning framework that exchanges models by exchanging **seed-projection pairs**. However, our work aims to push the method of seed-projection pairs for model exchange to its limits. We show that our method enjoys numerous surprising benefits compared to its predecessors. Moreover, we extend the experiments to models of larger scales and account for vision models also.

### 2.3 DATA HETEROGENEITY, BYZANTINE ATTACKS, AND COMPRESSION IN FL

Data heterogeneity is a critical concern in federated learning Ye et al. (2023) where each user holds inconsistent shards that do not represent the overall data distribution well, causing divergent updates and undermining training effectiveness. This can cause the global model to converge to suboptima with potential performance loss Karimireddy et al. (2020); Li et al. (2020). Efforts to address this

challenge under an FO setting are ongoing Qu et al. (2022); Fang et al. (2023); Jiang et al. (2023b); Chen et al. (2024b). Notably, Li et al. (2019) proposes to do a one-bit element-wise compression on the model weights to simultaneously promote data heterogeneity resilience and reduce communication load, pushing elementwise compression to its limit. However, the communication overhead still scales to the parameter size, hindering integration with large models.

Additionally, FL performance can be degraded by Byzantine clients who maliciously alter their data or models Fang et al. (2020), necessitating robust FL algorithms So et al. (2020); Tian et al. (2022). Within the context of zeroth-order (ZO) optimization, CYBER-0 Delgado Neto et al. (2024) marks an initial attempt by using trimmed mean aggregation to enhance the Byzantine resilience of ZO-based FL. Various aggregation methods have been proposed to improve FO-based FL resilience against such attacks Blanchard et al. (2017); Yin et al. (2018); Alistarh et al. (2018); So et al. (2020). Allouah et al. (2023) explores a joint defense method against data heterogeneity and Byzantine attacks. However, the communication load is not reduced and will be prohibitively high. Notably, Lang et al. (2023a;b) introduces a Byzantine resilient compressed aggregation method for FO-based FL systems where the communication overhead is reduced to 1 bit per step using a nested lattice coding with strict privacy guarantees, demonstrating that well-designed lossy compression can induce strong robustness without obviously compromising the performance of FL systems.

However, we notice that most efforts addressing this issue are separately doing **accurate** gradient estimation followed by **lossy** compression, leading to potentially unnecessary computational loads, as the compression eventually negates the costly effort of acquiring an accurate gradient estimation in FO-based methods. Motivated by this, we envisage a more integrated and efficient framework that runs on gradient estimation that is less accurate but attainable and communicable with much lower overheads with marginal performance loss. This marks a difference in rationale between our work and conventional methods addressing data heterogeneity and Byzantine attacks by compression.

## 3   *FeedSign*: ALGORITHM DESIGN AND CONVERGENCE ANALYSIS

---

**Algorithm 1:** FL with model exchange using seed-projection pairs

---

**Input:** Initialized model parameters $\boldsymbol{w}_0 \in \mathbb{R}^d$, loss function $\mathcal{L} : \mathbb{R}^d \to \mathbb{R}$, step budget $T$, client index set
    $k \in \mathcal{K} = \{1, \dots, K\}$, collections of client datasets $\{\mathcal{D}_k\}_{k \in \mathcal{K}}$, perturbation scale $\mu$, learning rate $\eta$
**Output:** Trained model parameters $\boldsymbol{w}_T$
Clients initialize model to $\boldsymbol{w}_0$
**for** $t = 1, \dots, T$ **do**
    PS **broadcasts seed** $s_t$   `// only for` **`FeedSign`**
    **for** $k = 1, \dots, K$ **do**
        `// clients do in parallel`
        Client update local model according to Equation 3 if receives a **projection broadcast**
        Client sample PRNG seed $s_{t,k}$   `// only for` **`ZO-FedSGD`**
        Client set PRNG seed to $s_t$   `// only for` **`FeedSign`**
        Client compute $p_k$ according to Equation 2
        Client send $p_k$ to PS
        Client send $s_{t,k}$ to PS   `// only for` **`ZO-FedSGD`**
    **end for**
    PS collects $p_1, \dots, p_k$ calculate projection $f(p_1, \dots, p_k)$ according to Equation 4
    PS **broadcasts projection** $f(p_1, \dots, p_k)$
**end for**

---

### 3.1   ALGORITHM DESIGN FOR MODEL EXCHANGE USING SEED-PROJECTION PAIRS

For transformer-based large models, *training* using gradient-based methods usually takes up 12 times of memory that is required by *inference* Malladi et al. (2023). The excessive demand for memory is due to complex operations of gradient backpropagation Rumelhart et al. (1986). One effective method of depriving the extra demand is using backpropagation-free optimizers, as is applied in *FwdLLM* Xu et al. (2024) and *FedKSeed* Qin et al. (2023). The methods proposed by these two works will be referred to as *ZO-FedSGD* for convenience. A brief description of the whole ZO-based FL is as Algorithm 1. Missing proofs can be found in the Appendices.

**Definition 1** (Client Update). *Consider a batch $\mathcal{B}$ from the dataset $\mathcal{D}$, a DL model whose parameter vector is $\boldsymbol{w} \in \mathbb{R}^d$, and a loss function $\mathcal{L}$, the applied ZO gradient estimator SPSA (Simultaneous*

*Perturbation Stochastic Approximation) estimates the gradient as*

$$p = \frac{\mathcal{L}(\boldsymbol{w} + \mu\boldsymbol{z}, \mathcal{B}) - \mathcal{L}(\boldsymbol{w} - \mu\boldsymbol{z}, \mathcal{B})}{2\mu}, \tag{1}$$

*where $\boldsymbol{z} \sim \mathcal{N}(\boldsymbol{0}, \boldsymbol{I}_d)$ is a Gaussian vector and $\mu$ is the perturbation scale and $p$ is the gradient projection.*

Given Definition 1, we apply a different update rule for *FeedSign* elaborated as

$$(\boldsymbol{ZO\text{-}FedSGD}) \quad \hat{\nabla}_{\boldsymbol{w}}\mathcal{L}(\boldsymbol{w}, \mathcal{B}) = p\boldsymbol{z}; \quad (\boldsymbol{FeedSign}) \quad \hat{\nabla}_{\boldsymbol{w}}\mathcal{L}(\boldsymbol{w}, \mathcal{B}) = \text{Sign}(p)\boldsymbol{z}. \tag{2}$$

The gradient estimate generated by SPSA can be broken into two parts, the random vector $\boldsymbol{z}$ and its corresponding gradient projection $p$. As a result, only the *seed* and the *gradient projection* are needed to be sent to PS. Different from Lang et al. (2023a;b), the shared PRNG is used directly to spawn the random vector $\boldsymbol{z}$ after which the devices scale it by $p$ to perfectly reconstruct the gradient estimation[1], which is done as follows:

**Definition 2** (Update Aggregation). *The global model of FL updates with learning rate $\eta$ under the following rule:*

$$(at\ client) \quad \boldsymbol{w} \leftarrow \boldsymbol{w} - f(p_1, \ldots, p_k)\eta\boldsymbol{z}, \tag{3}$$

*where*

$$(\textbf{ZO-FedSGD}) \quad f(p_1, \ldots, p_k) = \frac{1}{K}\sum_{k=1}^{K} p_k; \quad (\textbf{FeedSign}) \quad f(p_1, \ldots, p_k) = Sign\left(\sum_{k=1}^{K} \frac{p_k}{|p_k|}\right) \tag{4}$$

*with $K$ participating clients.*

*Remark* 1. Different from ZO-FedSGD, *FeedSign* always assumes that in a communication round, all clients perturb its model in the same direction for gradient estimation. Also, *FeedSign* left the sampling of random seeds to the PS and discards the amplitude of the gradient projection, whereas the PS uses a majority vote to determine whether the model should march or retreat a step of fixed size along the designated direction, allowing a **1-bit** per step communication overhead for *FeedSign*.

As a result, a comparison of communication overhead between *FeedSign* and the baseline ZO-FedSGD is elaborated as follows, assuming that only one random seed is explored per step.

$$
\begin{aligned}
(\boldsymbol{ZO\text{-}FedSGD}) \quad & \underbrace{1}_{\text{number of random seed}} \times \underbrace{32}_{\text{float number as gradient projection}} && \text{bits} \\
& + \underbrace{1}_{\text{number of random seed}} \times \underbrace{32}_{\text{long integer as random seed}} && \text{bits} \quad = 64 \text{ bits}, \\
(\boldsymbol{FeedSign}) \quad & \underbrace{1}_{\text{number of random seed}} \times \underbrace{1}_{\text{float number as gradient projection}} && \text{bit} \quad = 1 \text{ bit}.
\end{aligned} \tag{5}
$$

## 3.2 CONVERGENCE ANALYSIS

Some well-adopted assumptions are needed to facilitate the convergence analysis.

**Assumption 1** (*L*-smooth, Bottou et al. (2018)). *For any unbiased gradient estimate $\boldsymbol{g}(\boldsymbol{w})$ with finite second momentum, it satisfies*

$$\mathcal{L}(\boldsymbol{w}_{t+1}) \leq \mathcal{L}(\boldsymbol{w}_t) + \langle \nabla\mathcal{L}(\boldsymbol{w}_t), \boldsymbol{w}_{t+1} - \boldsymbol{w}_t \rangle + \frac{L}{2}\|\boldsymbol{w}_{t+1} - \boldsymbol{w}_t\|_2^2. \tag{6}$$

---

[1]The premise that all clients participating in the FL system share a PRNG is fulfilled since DL algorithms are often involved with random operations hence mainstream DL frameworks like Tensorflow and PyTorch provide PRNGs in consideration of reproducibility in random operations in training DL models. The default choice of PRNG is Philox Salmon et al. (2011) in Tensorflow and PyTorch, a deterministic algorithm with a guarantee that a fixed seed will always produce the same random integer stream while satisfying some statistical constraints.

**Assumption 2** (Local $r$-Effective Rank, Malladi et al. (2023), Assumption 1). *There is a matrix $\boldsymbol{H}(\boldsymbol{w}_t) \preceq \ell \boldsymbol{I}_d$ such that with $G(\boldsymbol{w}_t) = \max_{(\boldsymbol{x}, \boldsymbol{y}) \in \mathcal{D}} \|\nabla \mathcal{L}(\boldsymbol{w}_t, (\boldsymbol{x}, \boldsymbol{y}))\|_2$,*

*1. For all $\boldsymbol{w}$ such that $\|\boldsymbol{w} - \boldsymbol{w}_t\| \leq \eta d G(\boldsymbol{w}_t)$, we have $\nabla^2 \mathcal{L}(\boldsymbol{w}) \preceq \boldsymbol{H}(\boldsymbol{w}_t)$.*

*2. The effective rank of $\boldsymbol{H}(\boldsymbol{w}_t)$, i.e., $tr(\boldsymbol{H}(\boldsymbol{w}))/\|\boldsymbol{H}(\boldsymbol{w}_t)\|_{op}$, is at most $r$.*

**Assumption 3** (Unbiased Gradient Estimator with Bounded Data Heterogeneity). *The gradient estimator in Definition 1 is unbiased, specifically,*

$$\mathbb{E}_{\mathcal{B}}[\hat{\nabla}\mathcal{L}_k(\boldsymbol{w}, \mathcal{B})] = \nabla \mathcal{L}_k(\boldsymbol{w}), \tag{7}$$

$$\mathbb{E}_{\mathcal{B}}\left[\|\hat{\nabla}\mathcal{L}_k(\boldsymbol{w}, \mathcal{B})\|_2^2\right] \leq c_g \|\nabla \mathcal{L}_k(\boldsymbol{w})\|_2^2 + \frac{\sigma_g^2}{KB}\mathbb{V}[\nabla\mathcal{L}(\boldsymbol{w})], \tag{8}$$

$$\mathbb{E}_k\left[\|\nabla\mathcal{L}_k(\boldsymbol{w}) - \nabla\mathcal{L}(\boldsymbol{w})\|_2^2\right] \leq c_h \|\nabla\mathcal{L}(\boldsymbol{w})\|_2^2 + \sigma_h^2. \tag{9}$$

**Assumption 4** (Polyak-Łojaciewicz Inequality, Polyak (1964); Karimi et al. (2016); Malladi et al. (2023)). *Assume $\mathcal{L}^* := \min_{\boldsymbol{w} \in \mathbb{R}^d} \mathcal{L}(\boldsymbol{w}) > -\infty$, then there is a constant $\alpha > 0$ such that for any $\boldsymbol{w} \in \mathbb{R}^d$, $\mathcal{L}(\boldsymbol{w})$ satisfies*

$$\|\nabla\mathcal{L}(\boldsymbol{w})\|_2^2 \geq 2\delta(\mathcal{L}(\boldsymbol{w}) - \mathcal{L}^*), \quad \mathbb{V}[\nabla\mathcal{L}(\boldsymbol{w})] \leq 2\alpha(\mathcal{L}(\boldsymbol{w}) - \mathcal{L}^*). \tag{10}$$

**Assumption 5** (Sign Reversing Probability). *The gradient estimator has a reversed sign with the true gradient with probability $p_t$. Specifically, the expectation of Equation 2 satisfies*

$$p_t := Prob[p\bar{p} < 0], \quad \mathbb{E}_{\mathcal{B}}\left[\hat{\nabla}\mathcal{L}(\boldsymbol{w}, \mathcal{B})\right] = \bar{p}\boldsymbol{z}. \tag{11}$$

**Theorem 1** (Global Convergence for FedSGD, ZO-FedSGD, and *FeedSign*). *Given all assumption including Assumptions 1-5 satisfied, with corresponding conditions met, after*

$$t = A \log \frac{\mathcal{L}(\boldsymbol{w}_0) - \mathcal{L}^* - \tilde{C}}{\epsilon} \tag{12}$$

*steps, we will have the gap between the expected loss $\mathbb{E}[\mathcal{L}(\boldsymbol{w}_t)]$ and its possible lowest value $\mathcal{L}^* + \tilde{C}$ smaller than $\epsilon$ with*

$$(\textbf{FedSGD}) \quad A = \left(2\delta\eta - L\delta\eta^2 c_g(1 + c_h) - \frac{L\alpha\sigma_g^2\eta^2}{KB}\right), \qquad C = \frac{Lc_g\sigma_h^2\eta^2}{2}; \tag{13}$$

$$(\textbf{ZO-FedSGD}) \quad A = \left(2\delta\eta - L\zeta\delta\eta^2 c_g(1 + c_h) - \frac{L\zeta\alpha\sigma_g^2\eta^2}{KB}\right), \quad C = \frac{L\zeta c_g\sigma_h^2\eta^2}{2}; \tag{14}$$

$$(\textbf{FeedSign}) \quad A = 2\delta\eta(1 - 2\max_t p_t)\sqrt{\frac{2}{\pi}}, \qquad C = \frac{L\eta^2}{2}, \tag{15}$$

*where $\tilde{C} = C/A$ is the error floor with $0 < A < 1$ and $C > 0$, and $\zeta$ is a low-rank factor of the pre-trained model.*

*Remark* 2. **Convergence Rate Comparison.** Theorem 1 above shows that under a *FedSGD*-style setting, both *ZO-FedSGD* and *FeedSign* converges at an exponential rate $\mathcal{O}(e^{-t})$, the same rate as in FO methods can attain in big $\mathcal{O}$ notation. Notably, *ZO-FedSGD* differs from *FedSGD* to only a term characterizing the low-rank property of the pre-trained model $\zeta \sim \mathcal{O}(r)$. The parameter $r$ is found to be small compared to model size $d$ in well-trained DL models as reported in Papyan (2020); Ghorbani et al. (2019); Yao et al. (2020); Sagun et al. (2017); Wu et al. (2020).

*Remark* 3. **Data Heterogeneity Resilience.** It is observed that the error floor of *ZO-FedSGD* scales to the data heterogeneity parameters $c_g$ and $\sigma_h$ while that of *FeedSign* is independent of them. As a result, under an ideal iid case, the error floor vanishes with $\sigma_h = 0$ and $c_g \ll \infty$, but grows under high data heterogeneity. Contrarily, the error floor of *FeedSign* is fixed. In summary, we trade for more resilience against data heterogeneity at the cost of having a fixed but small error floor in *FeedSign*.

Table 1: Results on RoBERTa-large over language tasks. The best results obtained using federated ZO optimization is **bolded**, and the metric gap to that of the FO method is reported in the rightmost column. More results in Appendices.

| Task | SST-2 | SST-5 | SNLI | MNLI | RTE | TREC | Gap |
|---|---|---|---|---|---|---|---|
| Type | — sentiment — | | - natural language inference - | | | – topic – | |
| Zero-shot | 79.0 | 35.5 | 50.2 | 48.8 | 51.4 | 32.0 | – |
| | | | $k = 16$ | | | | |
| FO | 91.8 | 47.5 | 77.5 | 70.0 | 66.4 | 85.0 | – |
| MeZO | 90.5 | 45.5 | 68.5 | 58.7 | 64.0 | 76.9 | -5.6 |
| ZO-FedSGD | **89.7** | **46.8** | 63.1 | **60.5** | 63.1 | 70.0 | -7.5 |
| *FeedSign* | 88.9 | 45.0 | **69.7** | 59.7 | **65.3** | **75.6** | **-5.8** |
| | | | $k = 512$ | | | | |
| FO | 93.9 | 55.9 | 88.7 | 84.4 | 82.7 | 97.3 | – |
| MeZO | 93.3 | 53.2 | 83.0 | 78.3 | 78.6 | 94.3 | -3.7 |
| ZO-FedSGD | **93.0** | **52.0** | **84.9** | 74.8 | **76.8** | **94.4** | **-4.5** |
| *FeedSign* | 92.6 | 50.4 | 83.1 | **76.0** | 74.3 | 93.0 | -5.5 |

*Remark* 4. **Byzantine Resilience.** Nevertheless, $p_t$ is a key factor influencing the performance of *FeedSign*. It is noticed that for *ZO-FedSGD* and *FeedSign*, any attacks altering gradient estimation boils down to altering the gradient projection due to the deterministic nature of PRNG. While in *ZO-FedSGD*, clients have some degree of freedom to enact their strategies of attack hence being more unpredictable, the most effective method of damaging convergence of FFT due to the binary voting scheme in *FeedSign* is to always send a reversed sign to PS. The analytic characterization of its impact is succinct.

**Proposition 1** (Reversed Sign Probability with Byzantine Clients). *The batch gradient estimator* $\hat{\nabla}\mathcal{L}_k(\boldsymbol{w}_t, \mathcal{B})$ *will have a reversed sign to the true gradient* $\nabla\mathcal{L}$ *with a probability of*

$$p_t = p_{t,e} + p_{t,b} - p_{t,e}p_{t,b}, \tag{16}$$

*where* $p_{t,e}$ *is the inherent reversed sign probability due to batch gradient estimation error and* $p_{t,b}$ *is the proportion of Byzantine clients at step* $t$.

## 4 EXPERIMENTS

To validate the effectiveness of the proposed approach, we conducted extensive experiments across different tasks, data heterogeneity levels, and models of different types and sizes.

**Baselines.** To ensure consistency with previous research, we run the evaluation on RoBERTa-large, OPT-125M, and OPT-13B as is done in *MeZO*. We compare our method with standard FO methods (use backpropagation, takes up at least 6 times of memory), centralized ZO method *MeZO* Malladi et al. (2023) and *ZO-FedSGD* Xu et al. (2024); Qin et al. (2023). We kept the number of total perturbations consistent with that adopted in *MeZO*. As a result, the number of elapsed steps of *MeZO* is $K$ times that of *ZO-FedSGD* and *FeedSign*. We run both of the algorithms for the same number of steps, so the total communication overhead of *FeedSign* is $1/64$ of that of *ZO-FedSGD*.

### 4.1 MAIN RESULTS IN GENERAL SETTINGS

**Language models.** As is done in *MeZO*, we run few-shot learning for classification tasks on RoBERTa-large under two different settings, $k = 16$ and $k = 512$ samples per category, and general fine-tuning on OPT models. We employ test accuracy as the metric for classification and multiple-choice tasks and F1 score for generation tasks. Results are reported in Table 1 and 2, respectively.

It can be observed that *FeedSign* manifests no obvious performance gap to *MeZO* despite being a federated method with gradient projections of the lowest numerical resolution. This property scales up to an OPT model with 13B parameters. The largest metric gap between *MeZO* and centralized FO in the experiments is $-9.4\%$.

When fine-tuning RoBERTa-large using FO method and *FeedSign*, the mean performance gaps across the 6 few-shot learning tasks with $k = 16$ and $k = 512$ are $-5.8\%$ and $-5.5\%$, respectively. Besides, fine-tuning OPT-13B yields a mean gap of $-6.0\%$ over 11 tasks, narrower than that

Table 2: Main results on OPT-13B over language tasks. The highest metric obtained using federated ZO optimization is **bolded**, and the metric gap to that of the FO method is reported in the rightmost column.

| Task Type | SST-2 | RTE | CB | BoolQ | WSC | WIC | MultiRC | COPA | ReCoRD | SQuAD | DROP | Gap |
|---|---|---|---|---|---|---|---|---|---|---|---|---|
| | — classification — | | | | | | | — multiple choice – | | — generation — | | |
| Zero-shot | 58.8 | 59.6 | 46.4 | 59.0 | 38.5 | 55.0 | 46.9 | 80.0 | 81.2 | 46.2 | 14.6 | – |
| FO | 92.0 | 70.8 | 83.9 | 77.1 | 63.5 | 70.1 | 71.1 | 79.0 | 74.1 | 84.9 | 31.3 | – |
| MeZO | 91.4 | 66.1 | 67.9 | 67.6 | 63.5 | 61.1 | 60.1 | 88.0 | 81.7 | 84.7 | 30.9 | -3.1 |
| *ZO-FedSGD* | 84.7 | 60.2 | **67.8** | 64.1 | 52.8 | 55.3 | 54.1 | 84.0 | **81.7** | 76.1 | **29.4** | -7.9 |
| *FeedSign* | **87.7** | **62.0** | 67.8 | **64.5** | **60.5** | **55.7** | **57.3** | **88.0** | 81.7 | **77.6** | 28.5 | **-6.0** |

Table 3: Main results on OPT-125M over language models with iid with different sizes of client pool.

| Task | K | SST-2 | RTE | CB | BoolQ | WSC | WIC | MultiRC | COPA | ReCoRD | SQuAD | DROP |
|---|---|---|---|---|---|---|---|---|---|---|---|---|
| Zero-shot | - | 51.2 | 53.0 | 48.2 | 41.5 | 37.5 | 51.2 | 49.7 | 69.0 | 51.7 | 9.5 | 4.4 |
| MeZO | - | 82.2 | 55.9 | 67.8 | 61.0 | 59.6 | 51.0 | 53.3 | 68.0 | 47.1 | 44.1 | 15.2 |
| *ZO-FedSGD* | 5 | **84.4** | 57.0 | **67.8** | 59.1 | **57.6** | 49.5 | 51.2 | **60.0** | 48.7 | **46.9** | 16.1 |
| | 25 | 84.2 | 53.0 | 66.0 | 59.9 | **55.7** | 51.4 | 46.6 | **68.0** | 49.2 | 34.0 | 12.5 |
| *FeedSign* | 5 | 84.2 | **59.9** | **67.8** | **61.0** | 46.1 | **57.6** | **62.3** | 59.0 | 45.7 | **46.9** | **18.2** |
| | 25 | **85.0** | **60.6** | **67.8** | **60.6** | 51.9 | **56.5** | **55.4** | 63.0 | **49.3** | 45.9 | **16.0** |

of the ZO baseline. We observe that both *ZO-FedSGD* and *FeedSign* reach test accuracy pardonably lower but comparable to *ZO-FedSGD*.

We report the performance of *FeedSign* and *ZO-FedSGD* in Table 3 with client pool size $K = 5$ and 25. It can be observed that both of the methods maintain performance with a larger client pool size.

**Vision models.** Table 4 reports the test accuracy of *ZO-FedSGD* and *FeedSign* on CIFAR-10 and CIFAR-100. We download a pre-trained model checkpoint [2] and replace the classifier layer with a random initialized layer. It is shown that *FeedSign* attains a test accuracy of $91.7\%$ in only $2 \times 10^4$ steps with the support of a pre-trained model, faster than the ZO-based training SOTA Chen et al. (2023); Zhang et al. (2024) to the best of our knowledge with a much lesser number of steps.

Table 4: Results on ViT-large FFT.

| Dataset | CIFAR-10 | CIFAR-100 |
|---|---|---|
| ZO-trained SOTA | 86.5 | 34.2 |
| *ZO-FedSGD* | **94.0** | **62.7** |
| *FeedSign* | 91.7 | 45.3 |

### 4.2 DATA HETEROGENEITY RESILIENCE

**Settings.** A common approach to generating heterogeneous splits of a dataset is to have the number of samples from a class $c$ being proportional to $p_c \sim \text{Dirichlet}(\beta)$ for a client where $\alpha$ is a controlling parameter. Smaller $\beta$ will result in larger data heterogeneity among client datasets Vahidian et al. (2023).

**Language models.** Table 5 reports the test metric of *ZO-FedSGD* and *FeedSign*. We observe a drastic drop in test metrics through all tasks, confirming FL's vulnerability to data heterogeneity. However, it is clear that *FeedSign* outperforms *ZO-FedSGD* on most of the entries.

**Vision models.** We conduct a full-parameter FFT on a ResNet-18 checkpoint[3]. We observe that although *ZO-FedSGD* outperforms *FeedSign* on iid data, *FeedSign* turns the tide under high data heterogeneity. This affirms the theoretically implied data heterogeneity robustness of *FeedSign*.

However, we also notice that for the last-layer FFT on a ViT-large model, although *FeedSign* performs closely to *ZO-FedSGD*, it cannot outperform. We infer that this could be accounted for by the good feature extraction ability of ViT models.

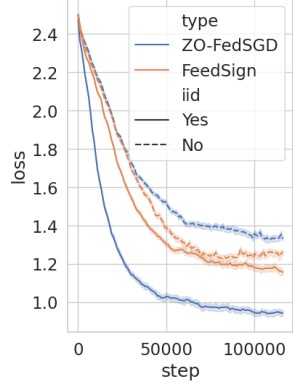

Figure 2: Loss curve versus number of steps elapsed.

---

[2]from `https://huggingface.co/google/vit-base-patch16-224`
[3]from `https://huggingface.co/microsoft/resnet-18`

Table 5: Main results on OPT-125M over language models with iid and non-iid data. We **bolded** the higher result within *FeedSign* and *ZO-FedSGD*.

| Task | SST-2 | RTE | CB | BoolQ | WSC | WIC | MultiRC |
|---|---|---|---|---|---|---|---|
| Zero-shot | 51.2 | 53.0 | 48.2 | 41.5 | 37.5 | 51.2 | 49.7 |
| *ZO-FedSGD* | 82.3 | 50.9 | **69.6** | 59.0 | **51.9** | 50.7 | 54.4 |
| ***FeedSign*** | **84.2** | **54.5** | 67.8 | **60.2** | 49.0 | **53.4** | **56.0** |
| *ZO-FedSGD*, $\beta = 1.0$ | 70.7 | **47.2** | 64.2 | 40.6 | **36.5** | 50.3 | **44.6** |
| ***FeedSign***, $\beta = 1.0$ | **73.0** | **47.2** | **66.0** | **40.8** | **36.5** | 50.0 | 44.5 |

Table 6: Main results on OPT-125M over language models with a Byzantine attacker.

| Task | SST-2 | RTE | CB | BoolQ | WSC | WIC | MultiRC | COPA | ReCoRD | SQuAD | DROP |
|---|---|---|---|---|---|---|---|---|---|---|---|
| Type | | | | classification | | | | – multiple choice – | | — generation — | |
| Zero-shot | 51.2 | 53.0 | 48.2 | 41.5 | 37.5 | 51.2 | 49.7 | 69.0 | 51.7 | 9.5 | 4.4 |
| *ZO-FedSGD* | 80.0 | **54.5** | **67.8** | **60.7** | 44.2 | 52.3 | 52.4 | 62.0 | 48.7 | 34.7 | 11.7 |
| ***FeedSign*** | **83.4** | 54.1 | 66.0 | 58.6 | **45.1** | **53.2** | **54.7** | **67.0** | **49.6** | **42.3** | **14.7** |

## 4.3 BYZANTINE RESILIENCE

**Settings.** We assume that there is one Byzantine client and 4 honest clients. The Byzantine client always transmits a random number as the gradient projection in *ZO-FedSGD*, and always transmits a reversed sign in *FeedSign*. All other settings are consistent with those listed in Section 4.2.

**Language models.** Table 6 reports the test metric of *ZO-FedSGD* and *FeedSign* with one of the clients as a Byzantine client. The test metric of *FeedSign* is higher than that of *ZO-FedSGD* with the largest gap of $+7.6\%$. It establishes that *FeedSign* expresses an inherent advantage in resisting Byzantine attacks.

**Image models.** Table 7 and Figure 3 report the test accuracy with one of the clients as a Byzantine client fine-tuning a ViT-large model. It can be observed that *ZO-FedSGD* is completely compromised with the Byzantine attack, while *FeedSign* maintains its performance.

## 5 DISCUSSIONS

With the performance of *FeedSign* well evaluated, we look further for some byproducts brought by the design of the framework.

## 5.1 EFFICIENT MODEL STORAGE AND SHARING

It is estimated that over $600,000$ models are stored in model sharing platforms like Huggingface, $90\%$ of them are fine-tuned models Ning et al. (2024). Frequently moving them results in PBs of monthly information transmission and storage demand. Notably, the platform can save only a small number of well-recognized checkpoints and save the *orbits*, which is the collection of seed-projection pairs elapsed from a checkpoint to fine-tuned models by using *FeedSign*-like methods, as shown in Figure 4. For example, for a fine-tuned OPT-13B model with $10,000$ fine-tune steps, 24GB of additional storage is required. However, the orbit generated by *FeedSign* will occupy less than 200 bytes of storage and guarantees perfect recovery of the fine-tuned model.

## 5.2 PARAMETER SERVERS CAN BE SMALL AND TASK AGNOSTIC

Figure 3: Loss and accuracy curve versus number of steps elapsed.

Table 7: Results on ViT-large FFT.

| CIFAR-100 | No attacker | One attacker |
|---|---|---|
| *ZO-FedSGD* | **62.7** | 10.9 |
| *FeedSign* | 45.3 | **40.8** |

| CIFAR-10 | No attacker | One attacker |
|---|---|---|
| *ZO-FedSGD* | **94.0** | 82.2 |
| *FeedSign* | 91.9 | **91.4** |

A byproduct of PS holding no actual DL model parameter of *FeedSign* is parameter security. This is because if operating *FedAvg* without special design like homomorphic encryption Liu et al. (2022), Mansouri et al. (2023), generic secure multiparty computation Burkhart et al. (2010), or additive masks So et al. (2021); Goryczka & Xiong (2015), the PS always knows the model parameters and hence has to be a legal holder of the final model. However, **not only data but also models are kept private and local** in FL systems featuring alike designs to *FeedSign*. In fact, according to Section 5.1, the PS can be a device that is too small to host the actual model. Moreover, conventional model-sharing platforms need to maintain large storage to store millions of models. However, with a *FeedSign*-like seed-projection pairs design as shown in Figure 4, the platform will not need to store the actual parameters, but only the orbits of elapsed seed-projection pairs during fine-tuning from some well-recognized checkpoints.

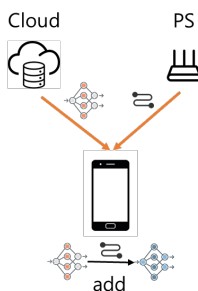

Figure 4: Orbit-based model sharing from a model agnostic third-party.

### 5.3 PRIVACY-CONVERGENCE TRADE-OFF

*FeedSign* can serve as an extremely memory-efficient framework that provides a strong privacy-convergence trade-off for different task requirements with a small modification on the aggregation rule.

**Definition 3** (Differentially Private Update Aggregation). *The global model of FL updates with learning rate $\eta$ under the following rule:*

$$(\textbf{\textit{DP-FeedSign}}) \quad \boldsymbol{w} \leftarrow \boldsymbol{w} - f_{DP}(p_1, \ldots, p_K)\eta\boldsymbol{z}. \tag{17}$$

*where $f_{DP}$ is a random variable with probability*

$$Prob(f_{DP} = 1) = p_+/(p_+ + p_-), \quad Prob(f_{DP} = -1) = p_-/(p_+ + p_-), \tag{18}$$

*where*

$$p_\pm = \exp\left(\frac{\epsilon q_\pm}{4}\right), \quad q_\pm = \sum_{k=1}^{K}\left(\frac{1}{2} \pm \frac{p_k}{|p_k|}\right) \tag{19}$$

*with $K$ participating clients.*

**Theorem 2** (Differential Privacy Guarantee). *Algorithm 1 with its update rule replaced as Definition 3 is $(\epsilon, 0)$-DP.*

*Remark 5.* By pushing $\epsilon$ to 0, we will have a stronger differential privacy (DP) guarantee, while the behavior of $f_{\text{DP}}$ will become more similar to Bernoulli(0.5). This will result in $p_t$ in Theorem 1 approaching $1/2$, slowing down the convergence of *FeedSign*.

*Remark 6.* Like Tang et al. (2024), our DP follows a new mechanism by only privatizing the gradient projection while it differs by having a discrete output. This is based on the fact that with the seed being broadcast and all machines sharing the same PRNG, the only uncertainty about the gradient for a malicious user is the sign of the corresponding gradient projection.

## 6 CONCLUSION

We have presented a novel FFT framework *FeedSign* that can operate in an extremely deficient communication and memory budget. Facilitated by ZO optimization and shared PRNG, each client needs only to upload one bit to the PS and then download one bit as a global update direction metric in a step, and use up the memory amount equaling to that needed for inference. We conduct theoretical analysis implying that *FeedSign* has many interesting properties including different kinds of robustness. Extensive experiments have shown that reducing communication overhead affects the performance of *FeedSign* little. We discuss some surprising advantages brought by the minimalistic design of *FeedSign* and how it can facilitate better FL deployment.

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

Table 8: Descriptions of Symbols

| Symbol | Description |
|---|---|
| $B$ | Batch size |
| $\mathcal{B}$ | Data batch |
| $c_g, \sigma_g$ | Batch gradient estimation noise factor |
| $c_h, \sigma_h$ | Client-wise gradient estimation noise factor |
| $\mathcal{D}$ | Dataset |
| $d$ | Number of model parameters |
| $k$ | Index of clients |
| $K$ | Number of clients in an FL system |
| $\mathcal{L}$ | Loss function |
| $\mathcal{L}^*$ | Infimum value of loss function |
| $L$ | Smooth constant of the loss function |
| $\mathcal{N}$ | Gaussian distribution |
| $p_b$ | Probability of a client being a Byzantine client |
| $p_e$ | Inherent probability of a batch gradient estimate having a reversed sign |
| $p_t$ | Overall probability of a batch gradient estimate having a reversed sign |
| $s$ | Random seed |
| $T$ | Number of global steps (step budget) |
| $t$ | Index of global epochs (the number of total communication rounds) |
| $\boldsymbol{w}$ | Model parameter vector |
| $\alpha, \delta$ | Polyak-Łojaciewicz property constant |
| $\epsilon$ | Toleration threshold of the gap to error floor |
| $\zeta$ | Low-effective rank factor of the gradient estimator |
| $\eta$ | Learning rate |
| $\nabla$ | Gradient operator |
| $\mathbb{E}$ | Expectation operator |
| $\mathbb{V}$ | Variance operator |
| $\mathbb{R}^n$ | $n$-dimensional real number set |
| $\langle \cdot, \cdot \rangle$ | Inner product |
| $tr$ | Trace operator |
| $\| \cdot \|_{\text{op}}$ | Operator norm of matrices |
| $\mathcal{L}(\boldsymbol{w})$ | Loss function at model parameter $\boldsymbol{w}$ |
| $\mathcal{L}_k(\boldsymbol{w})$ | Loss function of client $k$ at model parameter $\boldsymbol{w}$ |
| $\hat{\mathcal{L}}_k(\boldsymbol{w}, \mathcal{B})$ | Loss function measured on data batch $\mathcal{B}$ at model parameter $\boldsymbol{w}$ on client $k$ |
| $\mathcal{N}(\boldsymbol{\mu}, \boldsymbol{\Sigma})$ | Multivariate Gaussian distribution with center $\boldsymbol{\mu}$ and $\boldsymbol{\Sigma}$ |

## A  DESCRIPTION OF SYMBOLS

Descriptions of the symbols used in this paper can be found in Table 8.

## B  DOES *FeedSign* HAVE BLIND SPOTS?

The problem is equal to "can the gradients generated by *FeedSign* span $\mathbb{R}^d$?" We provide a positive answer that will eliminate the possibility that the optimum lies outside of the reachable space of *FeedSign* as shown in Figure 5, granting the possibility for a model to reach optimum as follows.

**Proposition 2.** *Gradients of* FeedSign *span $\mathbb{R}^d$ with probability $1$ after $d$ steps.*

This conclusion follows directly that Gaussian random matrices are full rank with probability $1$ and applies to ZO-FedSGD as well.

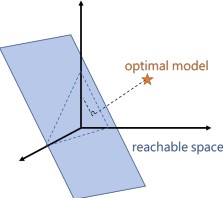

Figure 5: Optimum lying outside of the reachable space.

## C  PROOFS

Note that

$$\mathbb{V}[\nabla \mathcal{L}(\boldsymbol{w})] = B\left(\mathbb{E}\left[\nabla \mathcal{L}(\boldsymbol{w}; \mathcal{B}) \nabla \mathcal{L}(\boldsymbol{w}; \mathcal{B})^\top\right] - \nabla \mathcal{L}(\boldsymbol{w}) \nabla \mathcal{L}(\boldsymbol{w})^\top\right). \tag{20}$$

## C.1 PROOF TO THEOREM 1

For FedSGD, we have a well-known result

**Lemma 1** (Dimension-free Descent Lemma for FedSGD). *Given $\mathcal{L}(\boldsymbol{w})$ being a $L$-smooth function and $\hat{\nabla}\mathcal{L}(\boldsymbol{w}, \mathcal{B})$ an unbiased gradient estimator, the expected per-step loss descent can be bounded as follows:*

$$\mathbb{E}\left[\mathcal{L}(\boldsymbol{w}_{t+1})\right] \leq \mathcal{L}(\boldsymbol{w}_t) - \eta\|\nabla\mathcal{L}(\boldsymbol{w}_t)\|_2^2 + \frac{L\eta^2}{2}\mathbb{E}_{k,\mathcal{B}}\left[\|\nabla\mathcal{L}_k(\boldsymbol{w}_t, \mathcal{B})\|_2^2\right]. \tag{21}$$

This result follows combining the unbiasedness of the FO gradient estimator and Assumption 1.

For ZO-FedSGD, we will need the following lemma 2,

**Lemma 2** (Dimension-free Descent Lemma for ZO-FedSGD, Malladi et al. (2023)). *Given $\mathcal{L}(\boldsymbol{w})$ being a $L$-smooth function and $\hat{\nabla}\mathcal{L}(\boldsymbol{w}, \mathcal{B})$ an unbiased gradient estimator, the expected per-step loss descent can be bounded as follows:*

$$\mathbb{E}\left[\mathcal{L}(\boldsymbol{w}_{t+1})\right] \leq \mathcal{L}(\boldsymbol{w}_t) - \eta\|\nabla\mathcal{L}(\boldsymbol{w}_t)\|_2^2 + \frac{L\zeta\eta^2}{2}\mathbb{E}_{k,\mathcal{B}}\left[\|\nabla\mathcal{L}_k(\boldsymbol{w}_t, \mathcal{B})\|_2^2\right]. \tag{22}$$

*where*

$$\zeta = \frac{dr + d - 2}{n(d+2)} + 1 \tag{23}$$

*characterize the low-rank effect of the **gradient estimator**.*

*Remark* 7. Lemma 2 is the premise of successful ZO-based fine-tuning of large models. It can be observed that there is only an additional term in the quadratic term compared to that of the FO. It is the previous sense that SPSA-like algorithms result in a $\mathcal{O}(d)$ times larger gradient variance compared to FO methods Nemirovskij & Yudin (1983); Spall (1992); Jamieson et al. (2012); Oktay et al. (2020), prohibiting successful training of large models. However, Lemma 2 refined the bound and found that the gradient variance can be controlled by $\mathcal{O}(r)$, where $r$ is a loss landscape-related parameter known as local effective rank. The parameter is found to be small in well-trained DL models as reported in Papyan (2020); Ghorbani et al. (2019); Yao et al. (2020); Sagun et al. (2017); Wu et al. (2020).

*Proof.* We have

$$\mathbb{E}\left[\mathcal{L}(\boldsymbol{w}_{t+1})\right] \tag{24}$$

$$\leq \mathcal{L}(\boldsymbol{w}_t) - \eta\|\nabla\mathcal{L}(\boldsymbol{w}_t)\|_2^2 + \frac{L\zeta\eta^2}{2}\mathbb{E}_{k,\mathcal{B}}\left[\|\nabla\mathcal{L}_k(\boldsymbol{w}_t, \mathcal{B})\|_2^2\right] \tag{25}$$

$$\leq \mathcal{L}(\boldsymbol{w}_t) - \eta\|\nabla\mathcal{L}(\boldsymbol{w}_t)\|_2^2 + \frac{L\zeta\eta^2}{2}c_g(1+c_h)\|\nabla\mathcal{L}(\boldsymbol{w}_t)\|_2^2 + \frac{L\zeta\sigma_g^2\eta^2}{2KB}\mathbb{V}\left[\nabla\mathcal{L}(\boldsymbol{w}_t)\right] + \frac{L\zeta c_g\sigma_h^2\eta^2}{2} \tag{26}$$

$$\leq \mathcal{L}(\boldsymbol{w}_t) - \left(\eta - \frac{L\zeta\eta^2 c_g(1+c_h)}{2}\right)\|\nabla\mathcal{L}(\boldsymbol{w}_t)\|_2^2 + \frac{L\zeta\sigma_g^2\eta^2}{2KB}\mathbb{V}[\nabla\mathcal{L}(\boldsymbol{w}_t)] + \frac{L\zeta c_g\sigma_h^2\eta^2}{2} \tag{27}$$

$$\leq \mathcal{L}(\boldsymbol{w}_t) - \left(2\delta\eta - L\zeta\delta\eta^2 c_g(1+c_h) - \frac{L\zeta\alpha\sigma_g^2\eta^2}{KB}\right)(\mathcal{L}(\boldsymbol{w}_t) - \mathcal{L}^*) + \frac{L\zeta c_g\sigma_h^2\eta^2}{2}, \tag{28}$$

with a small enough $\eta$ satisfying

$$0 < \eta < 2/L\zeta c_g(1+c_h). \tag{29}$$

Substract $\mathcal{L}^*$ on both sides, then apply Assumption 4, we have

$$\mathbb{E}[\mathcal{L}(\boldsymbol{w}_{t+1})] - \mathcal{L}^* \leq \left(1 - \underbrace{\left(2\delta\eta - L\zeta\delta\eta^2 c_g(1+c_h) - \frac{L\zeta\alpha\sigma_g^2\eta^2}{KB}\right)}_{A_2}\right)(\mathcal{L}(\boldsymbol{w}_t) - \mathcal{L}^*) + \underbrace{\frac{L\zeta c_g\sigma_h^2\eta^2}{2}}_{C_2}. \tag{30}$$

With proper redistribution of the $C_1$ term, we have an error bound $\tilde{C}_2 = C_2/A_2$, and to reach an optimality gap smaller than $\epsilon$ will take

$$t = A_2 \log \frac{\mathcal{L}(\boldsymbol{w}_0) - \mathcal{L}^* - \tilde{C}_2}{\epsilon} \tag{31}$$

steps with $0 < A_2 < 1$.

We will have

$$\mathbb{E}[\mathcal{L}(\boldsymbol{w}_{t+1})] - \mathcal{L}^* \leq \left( 1 - \underbrace{\left( 2\delta\eta - L\delta\eta^2 c_g(1 + c_h) - \frac{L\alpha\sigma_g^2\eta^2}{KB} \right)}_{A_1} \right) (\mathcal{L}(\boldsymbol{w}_t) - \mathcal{L}^*) + \underbrace{\frac{Lc_g\sigma_h^2\eta^2}{2}}_{C_1} . \tag{32}$$

with a similar processing for FedSGD for its exponential convergence.

For *FeedSign*, since *FeedSign* does not guarantee unbiased gradient estimation, we will have to start from Assumption 1.

**Lemma 3** (Dimension-free Descent Lemma for *FeedSign*). *Given $\mathcal{L}(\boldsymbol{w})$ being a L-smooth function, the expected per-step loss descent can be bounded as follows:*

$$\mathbb{E}\left[\mathcal{L}(\boldsymbol{w}_{t+1})\right] \leq \mathcal{L}(\boldsymbol{w}_t) - \eta(1 - 2p_t)\sqrt{\frac{2}{\pi}}\|\nabla\mathcal{L}(\boldsymbol{w}_t)\|_2^2 + \frac{L\eta^2}{2}, \tag{33}$$

*where $\pi$ is the circumference ratio.*

*Proof.* Start from Assumption 1, with $\hat{\nabla}\mathcal{L}(\boldsymbol{w}, \mathcal{B})$ being the unbiased estimator used by *ZO-FedSGD*,

$$\mathbb{E}[\mathcal{L}(\boldsymbol{w}_{t+1})] \leq \mathcal{L}(\boldsymbol{w}_t) - \eta \left\langle \nabla\mathcal{L}(\boldsymbol{w}_t), \mathbb{E}_{\mathcal{B}}\left[\text{Sign}(\hat{\nabla}\mathcal{L}(\boldsymbol{w}_t, \mathcal{B}))\right] \right\rangle + \frac{L\eta^2}{2} \left\| \text{Sign}(\hat{\nabla}(\boldsymbol{w}_t, \mathcal{B})) \right\|_2^2 . \tag{34}$$

With Assumption 5, we have

$$\mathbb{E}_{\mathcal{B}}\left[\text{Sign}(\hat{\nabla}(\boldsymbol{w}_t, \mathcal{B}))\right] = \mathbb{E}_{\boldsymbol{z},\mathcal{B}}\left[\text{Sign}(\boldsymbol{z}^\top\nabla\mathcal{L}(\boldsymbol{w}_t, \mathcal{B}))\right] = (1 - 2p_t)\mathbb{E}_{\boldsymbol{z}}\left[\text{Sign}(\boldsymbol{z}^\top\nabla\mathcal{L}(\boldsymbol{w}_t))\right], \tag{35}$$

where the estimator can be elaborated as

$$\hat{\nabla}\mathcal{L}(\boldsymbol{w}_t, \mathcal{B}) = \boldsymbol{z}^\top\nabla\mathcal{L}(\boldsymbol{w}_t, \mathcal{B})\boldsymbol{z}. \tag{36}$$

Noticing that $\text{Sign}(x) = x/|x|$ and $\|\text{Sign}(\cdot)\| = 1$, we have the following

$$\mathbb{E}[\mathcal{L}(\boldsymbol{w}_{t+1})] \leq \mathcal{L}(\boldsymbol{w}_t) - \eta(1 - 2p_t)\mathbb{E}_{\boldsymbol{z}}\left\langle \nabla\mathcal{L}(\boldsymbol{w}_t), \frac{\boldsymbol{z}^\top\nabla\mathcal{L}(\boldsymbol{w}_t)}{|\boldsymbol{z}^\top\nabla\mathcal{L}(\boldsymbol{w}_t)|}\boldsymbol{z} \right\rangle + \frac{L\eta^2}{2} \tag{37}$$

$$= \mathcal{L}(\boldsymbol{w}_t) - \eta(1 - 2p_t)\mathbb{E}_{\boldsymbol{z}}\left[\frac{\boldsymbol{z}^\top\nabla\mathcal{L}(\boldsymbol{w}_t)\boldsymbol{z}^\top\nabla\mathcal{L}(\boldsymbol{w}_t)}{|\boldsymbol{z}^\top\nabla\mathcal{L}(\boldsymbol{w}_t)|}\right] + \frac{L\eta^2}{2} \tag{38}$$

$$= \mathcal{L}(\boldsymbol{w}_t) - \eta(1 - 2p_t)\mathbb{E}_{\boldsymbol{z}}\left[|\boldsymbol{z}^\top\nabla\mathcal{L}(\boldsymbol{w})|\right] + \frac{L\eta^2}{2}. \tag{39}$$

Since $\boldsymbol{z} \sim \mathcal{N}(\boldsymbol{0}, \boldsymbol{I}_d)$, $\boldsymbol{z}^\top\nabla\mathcal{L}(\boldsymbol{w}) \sim \mathcal{N}(0, \|\nabla\mathcal{L}(\boldsymbol{w}_t)\|_2^2)$, and the property of half-normal distribution tells that

$$\mathbb{E}_{\boldsymbol{z}}\left[|\boldsymbol{z}^\top\nabla\mathcal{L}(\boldsymbol{w})|\right] = \sqrt{\frac{2}{\pi}}\|\nabla\mathcal{L}(\boldsymbol{w}_t)\|_2^2. \tag{40}$$

arriving at Equation 33.

$\square$

The quadratic term of weight difference vanishes since *FeedSign* does not contain "amplitude" of the gradient projection, only a binary choice. Apply Assumption 4, we have

$$\mathbb{E}\left[\mathcal{L}(\boldsymbol{w}_{t+1})\right] \le \mathcal{L}(\boldsymbol{w}_t) - 2\eta\delta(1 - 2p_t)\sqrt{\frac{2}{\pi}}(\mathcal{L}(\boldsymbol{w}_t) - \mathcal{L}^*) + \frac{L\eta^2}{2}. \tag{41}$$

Subtract $\mathcal{L}^*$ on both sides, we have

$$\mathbb{E}[\mathcal{L}(\boldsymbol{w}_{t+1})] - \mathcal{L}^* \le \left(1 - \underbrace{2\eta\delta(1 - 2p_t)\sqrt{\frac{2}{\pi}}}_{A_3}\right)(\mathcal{L}(\boldsymbol{w}_t) - \mathcal{L}^*) + \underbrace{\frac{L\eta^2}{2}}_{C_3}, \tag{42}$$

with an error bound $\tilde{C}_3 = C_3/A_3$. To reach an optimality gap smaller than $\epsilon$ will take

$$t = A_3 \log \frac{\mathcal{L}(\boldsymbol{w}_0) - \mathcal{L}^* - \tilde{C}_3}{\epsilon} \tag{43}$$

steps with $0 < A_3 < 1$ and $C_3 > 0$. $\qquad\square$

## C.2 PROOF TO PROPOSITION 1

*Proof.* In *FeedSign*, assume at a particular point $\boldsymbol{w}_t$, the sign of the true gradient is $f_t = \nabla\mathcal{L}(\boldsymbol{w}_t)/|\mathcal{L}(\boldsymbol{w}_t)|$. We say that a client **successes** if it sends a correct sign to the PS, and **fails** otherwise. After local computation, honest clients always send the sign, and Byzantine clients always reverse the sign and then send it. Due to batch gradient noise, the probability of an honest fail is $p_{t,e}$ and an honest success is $1 - p_{t,e}$. Contradictorily, the probability of a Byzantine fail and Byzantine success is $1 - p_{t,e}$ and $p_{t,e}$, respectively. Assume the probability of a client being Byzantine is $p_{t,b}$.

During a vote, the number of fails is a random variable $V$ that follows a binomial distribution with

$$\mathbb{E}[V] = \frac{1}{2}K + (\frac{1}{2} - p_{t,e})(2p_{t,b} - 1)K, \tag{44}$$

$$\mathbb{V}[V] = (\frac{1}{4} - p_{t,e}^2). \tag{45}$$

The adjusted error rate with Byzantine clients will be $\mathbb{E}[V]/K$. $\qquad\square$

## C.3 PROOF TO THEOREM 2

*Proof.* Denote $\mathcal{F} := \{1, -1\}$, and $\boldsymbol{p} := (p_1, \ldots, p_K)$. Denote $\|\cdot, \cdot\|_1$ the Hamming distance of two vectors. Then for any $\boldsymbol{p} \in \mathcal{S}^K$ with $\|\boldsymbol{p}, \boldsymbol{p}'\|_1 \le 1$ and any $f \in \mathcal{F}$, denoting $\hat{f} := f_{\text{DP}}(\boldsymbol{p})$, $\hat{f}' := f_{\text{DP}}(\boldsymbol{p}')$,

$$\frac{\text{Prob}(\hat{f} = f)}{\text{Prob}(\hat{f}' = f)} = \frac{\exp(\epsilon q_{\hat{f}}/4)}{\exp(\epsilon q_{\hat{f}'}/4)} \frac{\exp(\epsilon q_{\hat{f}'}/4) + \exp(\epsilon q_{-\hat{f}'}/4)}{\exp(\epsilon q_{\hat{f}}/4) + \exp(\epsilon q_{-\hat{f}}/4)} \tag{46}$$

$$= \exp\left(\frac{\epsilon(q_{\hat{f}} - q_{\hat{f}'})}{4}\right) \frac{\exp(\epsilon(q_{\hat{f}} + 2)/4) + \exp(\epsilon(q_{-\hat{f}} + 2)/4)}{\exp(\epsilon q_{\hat{f}}/4) + \exp(\epsilon q_{-\hat{f}}/4)} \tag{47}$$

$$\le \exp\left(\frac{2\epsilon}{4}\right)\exp\left(\frac{2\epsilon}{4}\right) \frac{\exp(\epsilon q_{\hat{f}}/4) + \exp(\epsilon q_{-\hat{f}}/4)}{\exp(\epsilon q_{\hat{f}}/4) + \exp(\epsilon q_{-\hat{f}}/4)} \tag{48}$$

$$= \exp(\epsilon). \tag{49}$$

## C.4 PROOF FOR PROPOSITION 2

We begin the proof with a lemma regarding the property of Gaussian matrices.

**Lemma 4** (Gaussian Matrices are Full-rank with Proabability 1)**.** *Gaussian random matrices* $M \in \mathbb{R}^{q \times d}$, *whose elements are* $M_{ij} \sim \mathcal{N}(0, 1)$, $i = 1, \ldots, q, j = 1, \ldots, d$ *are full rank with probability 1.*

*Proof.* $M$ is not full rank if and only if $\det(MM^T) = 0$, which is equivalent to the existence of a polynomial $p : \mathbb{R}^{qd} \to \mathbb{R}$ such that $p(\vec{M}) = 0$, where $\vec{M} \in \mathbb{R}^{qd}$ stands the flattened $M$, a vector with all its elements collected non-repeatedly from $M$. Thus

$$\text{Prob}(M \text{ is not full rank}) = \int_{p(\vec{M})=0} \mathrm{d}F(\vec{M}), \tag{50}$$

where $F(\vec{M})$ is the cumulative distribution function (CDF) of $\vec{M}$. Note that $p$ is continuous, then $\mathcal{Z}(p) = \{\boldsymbol{x}|p(\boldsymbol{x}) = 0\}$ is Lebesgue measurable. Denote by $\mu(\cdot)$ the Lebesgue measure on $\mathbb{R}^{qd}$. By observing that $\mu(\mathcal{Z}(p)) = 0$ for any polynomial $p$, it implies that $M$ is full rank with probability 1.

We prove the above observation by induction. Suppose the conclusion holds for polynomials of order $n - 1$. Then for any polynomial of order $n$, we can write

$$p(\boldsymbol{x}, x_n) = \sum_{j=0}^{k} p_j(\boldsymbol{x}) x_n^j, \tag{51}$$

where $\boldsymbol{x} \in \mathbb{R}^{n-1}$ and at least $p_k$ is nontrivial. Denote $\tilde{\boldsymbol{x}} = (\boldsymbol{x}, x_n)$, then $\tilde{\boldsymbol{x}} \in \mathcal{Z}(p)$ requires one of the following disjoint events:

    A. $p_0(\boldsymbol{x}) = \cdots = p_k(\boldsymbol{x}) = 0$,

    B. $x_n$ solves $p_{\boldsymbol{x}}(t) = \sum_{j=0}^{k} p_j(\boldsymbol{x}) t^j$.

Let $A$ and $B$ be the respective sample set of the events. It is clear that $\mu(A) = 0$ by inductive hypothesis. The *fundamental theorem of algebra* shows that $B$ consists of at most $k$ points (atoms) and thereby $\mu(B) = 0$. Fubini's theorem guarantees that based on the above the set of $\boldsymbol{x}$ satisfying either $A$ or $B$ has a Lebesgue measure of zero. Then by $\mu(\mathcal{Z}(p)) = \mu(A \cup B) \leq \mu(A) + \mu(B) = 0$ and noticing the case of $n = 1$ is trivial establishes the induction. $\qquad\square$

With Lemma 4, it is clear that the gradients will span $\mathbb{R}^d$ with a sufficient number of steps.

$\qquad\square$

# D WHEN WILL *FeedSign* BE UNBIASED?

It is clear that *FeedSign* does not provide general unbiased gradient estimation. We will discuss a specific case where *FeedSign* is unbiased.

Assume the true gradient projection for a specific model is $p$, and the estimation yielded by *ZO-FedSGD* is $p_1$ and *FeedSign* $p_2$. Consider a noise $n$ with CDF $F(x)$ on $p$ due to batch gradient estimation, specifically, $p_1 = p + n$, then

$$\text{Prob}(p_2 = 1) = F(p), \tag{52}$$
$$\text{Prob}(p_2 = -1) = F(-p). \tag{53}$$

To make $p_2$ an unbiased estimator of $p$, we need

$$\mathbb{E}[p_2] = F(p) - F(-p) = p \tag{54}$$

for any $p$ in the support of $F(x)$. This result implies that *FeedSign* is unbiased only when

    1. the noise CDF is uniform on $[-1, 1]$,

    2. $p$ takes value on $[-1, 1]$ only.

This is an obviously unrealistic setting. However, there could be methods that distort the true distribution of $p$ and $n$ so that they behave similarly to the abovementioned case, making *FeedSign* an unbiased FL method. This could be a topic for future investigation.

# E    Hyperparameters

Table 9: Hyperparameters settings. *: See Table 10 for details.

| | $B$ | $T$ | $\eta$ | $K$ | $\mu$ | $\beta$ |
|---|---|---|---|---|---|---|
| Table 1 | 64 | $1 \times 10^5$ | $1 \times 10^{-6}$ for *ZO-FedSGD*, $5 \times 10^{-5}$ for *FeedSign* | 5 | $1 \times 10^{-3}$ | - |
| Table 2 | 16 | $2 \times 10^4$ | $1 \times 10^{-7}$ for *ZO-FedSGD*, $5 \times 10^{-6}$ for *FeedSign* | 5 | $1 \times 10^{-3}$ | - |
| Table 3 | 16 | * | $1 \times 10^{-7}$ for *ZO-FedSGD*, $5 \times 10^{-6}$ for *FeedSign* | * | $1 \times 10^{-3}$ | - |
| Table 4 | 16 | $2 \times 10^4$ for CIFAR-10 $6 \times 10^4$ for CIFAR-100 | $1 \times 10^{-3}$ | 5 | $1 \times 10^{-5}$ | - |
| Table 5 | 16 | $6 \times 10^4$ | $1 \times 10^{-7}$ for *ZO-FedSGD*, $5 \times 10^{-6}$ for *FeedSign* | 5 | $1 \times 10^{-3}$ | 1.0 |
| Table 6 | 16 | $6 \times 10^4$ | $1 \times 10^{-7}$ for *ZO-FedSGD*, $5 \times 10^{-6}$ for *FeedSign* | 5 | $1 \times 10^{-3}$ | - |
| Table 7, Figure 3 | 64 | $2 \times 10^4$ for CIFAR-10 $6 \times 10^4$ for CIFAR-100 | $1 \times 10^{-3}$ | 5 | $1 \times 10^{-5}$ | - |
| Figure 2 | 64 | $1.2 \times 10^5$ | $1 \times 10^{-4}$ | 25 | $1 \times 10^{-5}$ | 1.0 |

Table 10: Step budgets and numbers of perturbations used in Table 3.

| $K$ | Step budget | The number of perturbations $T$ |
|---|---|---|
| *MeZO* | $6 \times 10^4$ | $6 \times 10^4$ |
| 5 | $6 \times 10^4$ | $3 \times 10^5$ |
| 25 | $1.2 \times 10^4$ | $3 \times 10^5$ |

In Table 1 and Table 2, we kept the number of perturbations rather than step budget aligned to *MeZO*. In Table 3, we align the step budgets for $K = 5$ in our comparison with the centralized counterpart. However, since the computation complexity scales to the number of perturbations hence to client pool size $K$ also, we report the result of $K = 25$ with $1/5$ of step budgets. Other hyperparameters in Table 1-3 is set to be consistent to *MeZO* Malladi et al. (2023). In language model experiments, we set a larger learning rate $\eta$ since *FeedSign* may have a smaller gradient norm. We believe that this will be partially accounted for outperforming the reported figures in *MeZO* in several instances.

Additionally, we added a random multiplier following $1 + \mathcal{N}(0, 1)$ to gradient projection estimates of both *ZO-FedSGD* and *FeedSign* to simulate a high data heterogeneity with a high value of $c_g$ in Theorem 1 in Figure 2, apart from higher $\sigma_h$ caused by Dirichlet distributed client dataset.

# F    Examples of PyTorch PRNG

We include several snippets demonstrating the behavior of PyTorch PRNG and describe how it helps *FeedSign* cut down the communication overhead.

## F.1    Using PRNG for RGE

We repeatedly call `torch.randn_like` to spawn an identical random perturbation from a random seed.

```
def seed_perturb(self, seed, scale):
    torch.manual_seed(seed)

    for k, v in self.model.named_parameters():
        dv = torch.randn_like(v).to(v.device)
        v.data += dv * scale
```

### F.2 BRIEF DEMONSTRATION ON THE BEHAVIOR OF THE PYTORCH PRNG

We include a snippet as in Section F.3 to demonstrate the behavior of PRNG in PyTorch. We mainly execute the following operations in Python:

1. Set the random seed to $42$. (line 2)
2. Generate three Gaussian random arrays $a, b$, and $c$. (line 7 to 23)
3. Some operations that access the previously generated arrays. (line 24 to 26)
4. Reset the random seed to $42$. (line 27)
5. Some operations that access the previously generated arrays. (line 29 to 31)
6. Generate three Gaussian random arrays $a1, b1$, and $c1$ with shapes identical to $a, b$, and $c$, respectively. (line 32 to 51)

It is observed that:

1. Array $a, b$, and $c$ are identical to $a1, b1$, and $c1$, respectively, even with some operations that access the arrays.
2. Array $a$ and $c$ are not identical though they have the same shape. This is because between the two random array generations, `torch.manual_seed` is not called.

This above result is reproducible on the following four devices we have tested:

| Type | OS/CPU/GPU | Python | PyTorch | CUDA | cuDNN |
|---|---|---|---|---|---|
| Alienware x15 R1 | Windows 11
11th Gen Intel(R) Core(TM) i7-11800H @ 2.30GHz
1x NVIDIA GeForce RTX 3070 Laptop GPU | 3.10.6 | 2.3.1 | 12.4 | 8.0 |
| ASUS ESC8000 G4 | Linux 5.10.0, amd64
Intel(R) Xeon(R) Gold 6133 CPU @ 2.50GHz
6x NVIDIA GeForce RTX 3090 GPU | 3.10.13 | 2.3.0 | 12.1 | 8.9 |
| Inspur NF5488A5 | Linux 4.18.0, x86_64
AMD EPYC 7742 64-Core Processor
8x NVIDIA A100-SXM4-80GB | 3.11.9 | 2.2.0 | 12.1 | 8.9 |

## F.3 CODE BLOCK

```python
>>> import torch
>>> torch.manual_seed(42)
<torch._C.Generator object at 0x7fd2e469e890>
>>> a = torch.randn((5, 5))
>>> b = torch.randn((4, 6))
>>> c = torch.randn((5, 5))
>>> a
tensor([[ 1.9269,  1.4873,  0.9007, -2.1055,  0.6784],
        [-1.2345, -0.0431, -1.6047, -0.7521, -0.6866],
        [-0.4934,  0.2415, -1.1109,  0.0915, -2.3169],
        [-0.2168, -1.3847, -0.3957,  0.8034, -0.6216],
        [-0.5920, -0.0631, -0.8286,  0.3309, -1.5576]])
>>> b
tensor([[ 0.3211,  1.5736, -0.8455,  1.3123,  0.6872, -1.0892],
        [-0.3553, -0.9138,  0.8564,  2.2181,  0.5232,  0.3466],
        [-0.1973, -1.0546,  1.2780,  0.1453,  0.5238,  0.0566],
        [ 0.4263,  0.5750, -0.6417, -2.2064, -0.7508,  2.8140]])
>>> c
tensor([[-0.3387, -1.3407, -0.5854,  0.5362,  0.5246],
        [ 1.1412,  0.0516,  0.7281, -0.4816,  0.1877],
        [-0.3576, -0.3165,  0.5886, -0.8905,  0.4098],
        [-1.4570, -0.1023,  0.3499,  0.6173, -0.1693],
        [ 0.2332,  4.0356,  1.2795, -0.0127,  0.2408]])
>>> d = 3 + b[3, 2]
>>> d
tensor(2.3583)
>>> torch.manual_seed(42)
<torch._C.Generator object at 0x7fd2e469e890>
>>> e = 1 + b[2, 4]
>>> e
tensor(1.5238)
>>> a1 = torch.randn((5, 5))
>>> b1 = torch.randn((4, 6))
>>> c1 = torch.randn((5, 5))
>>> a1
tensor([[ 1.9269,  1.4873,  0.9007, -2.1055,  0.6784],
        [-1.2345, -0.0431, -1.6047, -0.7521, -0.6866],
        [-0.4934,  0.2415, -1.1109,  0.0915, -2.3169],
        [-0.2168, -1.3847, -0.3957,  0.8034, -0.6216],
        [-0.5920, -0.0631, -0.8286,  0.3309, -1.5576]])
>>> b1
tensor([[ 0.3211,  1.5736, -0.8455,  1.3123,  0.6872, -1.0892],
        [-0.3553, -0.9138,  0.8564,  2.2181,  0.5232,  0.3466],
        [-0.1973, -1.0546,  1.2780,  0.1453,  0.5238,  0.0566],
        [ 0.4263,  0.5750, -0.6417, -2.2064, -0.7508,  2.8140]])
>>> c1
tensor([[-0.3387, -1.3407, -0.5854,  0.5362,  0.5246],
        [ 1.1412,  0.0516,  0.7281, -0.4816,  0.1877],
        [-0.3576, -0.3165,  0.5886, -0.8905,  0.4098],
        [-1.4570, -0.1023,  0.3499,  0.6173, -0.1693],
        [ 0.2332,  4.0356,  1.2795, -0.0127,  0.2408]])
```

## G    TEST ACCURACY OF OTHER ZO METHODS ON CIFAR-10 DATASET

We list the test accuracy on CIFAR-10 dataset obtained by some previous ZO methods in Figure 6, including Pattern Search Chiang et al. (2022), Align-Ada Boopathy & Fiete (2022), LG-FG-A and FG-W Ren et al. (2022), and DeepZero Chen et al. (2023). To conduct a fair comparison, we present the test accuracy when *FeedSign* is run with only one client to simulate a centralized training manner since the listed baseline performances are all obtained under a centralized learning setting.

Notably, we are not able to compare our approach with DeepZero at the same scale. This is because in DeepZero, the authors used ResNeta-20, a version of ResNet tailored for images of small sizes ($3 \times 32 \times 32$, the standard CIFAR-10 size). Unfortunately, there are no available "pre-trained models" for models at this scale. In our implementation, we upsample the CIFAR-10 images to $3 \times 224 \times 224$ to adapt to the standard input shape of the standard version of ResNet-18. Moreover, DeepZero cannot scale up to ViT-large models due to prohibitively high computation overhead.

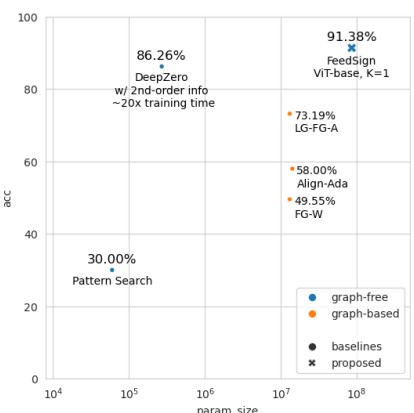

Figure 6: Test accuracy on CIFAR-10 dataset of some ZO baselines.

## H    IMPLEMENTATION DETAILS

### H.1    EXPERIMENTAL SETTINGS

To ensure consistency with previous research, we run the evaluation on RoBERTa-large, OPT-125M and OPT-13B as is done in *MeZO*. Additionally, we adapt the method to image models and run evaluations on ViT-base. Language models are run on a server equipped with 8 NVIDIA A100-80GB GPUs, and image models are run on a smaller server equipped with 6 NVIDIA GeForce RTX 3090 GPUs.

For hyperparameters, we follow the configuration of *MeZO* for language models and develop our own set of parameters for image models. The number of participating clients is set to $K = 5$. We set the random seed to $t$ at $t$-th step in *FeedSign*.

### H.2    MODEL PARAMETER UPDATE USING ZO METHODS

Two approaches can be used to update the model parameters in PyTorch:

1. Put the SPSA gradient estimate to the corresponding `param.grad`, and use the standard PyTorch `optimizer.step()` to update the model parameters.
2. Inplace subtracting the entries of the `state_dict` object of the PyTorch `model` by the SPSA gradient estimate.

Table 11: Memory consumption of RoBERTa-large when using batch size 16 with the MultiRC task. The reported memory does not include the cost of storing the model on the GPU.

| Task | *FeedSign*, Approach 2 Inference | *FeedSign*, Approach 1 Inference+optimizer | FO methods (common *FedAvg*) Backpropagation |
|---|---|---|---|
| Excess Memory (MB) | 327.50 | 830.66 | 24156.23 |

In terms of memory requirement, our methods is essentially same to what is done in Malladi et al. (2023). The benefit of the first approach is that it is compatible with optimizers like Adam and RM-

SProp provided by PyTorch, and the drawback is that those optimizers often use momentum, resulting in 2x or 3x times the memory consumption compared to model inference, but still significantly smaller than that of the memory consumption of FO methods. The second approach consumes the exact same amount of memory compared to inference, however, it can result in slower convergence. We use the first approach for image models and the second for language models.

### H.3 IMPLEMENTING *FedSGD* OVER LLM

A common way of building an FL system simulation is maintaining $K + 1$ model instances in the memory ($K$ as the clients, one as the PS). However, the largest experiment we have carried out is FO full-parameter fine-tuning OPT-13B with *FedAvg* over $K = 5$ clients. Fine-tuning an instance of an OPT-13B model costs 316 GB (4xA100 GPU) GPU memory, and 6 of them is unaffordable for us. So we make use of the behavior of the auto-differentiation of PyTorch to detour the problem.

In a nutshell, we simulate only the global model on a virtual PS in the memory, and the local updates from different clients are accumulated to `param.grad` by calling `loss.backward()` for $K$ times, each time on the `loss` computed from a corresponding client. The `loss.backward()` implicitly simulates three steps in a communication round: 1) clients computing local updates, 2) clients sending local updates to the PS, 3) PS aggregates the local gradients. Next, a call of `optimizer.step()` subtracts the parameters by the corresponding entries of `param.grad`, simulating the global model marching a step along the gradient direction and broadcasting the updated model. In this way, we simulate the *FedSGD* over $K$ clients using only 1x inference memory.

We illustrate the process using the following snippet.

```
for t in range(T):
    optimizer.zero_grad()
    for c in range(C):
        '''
        sample a batch from its private dataset
        calculate local loss
        '''
        loss.backward()      # accumulate local gradients to global model
    optimizer.step()
```

### H.4 MORE RESULTS

In this section, we will illustrate more experimental details.

**Main results.** We additionally compare two *ZO-FedAvg* baselines with different aggregation frequency, $l = 5$ steps and $l = bn$ steps, where $bn$ is the number of batches in the dataset. Table 12 provides supplemental results of Table 1. In fine-tuning an RoBERTa-large model, compared to the best federated ZO method, the performance gaps are within $-2\%$ in 10 of the 12 entries, including 2 where *FeedSign* outperforms the baseline methods. Table 13 provides supplemental results of Table 2. In fine-tuning an OPT-13B model, compared to the best federated ZO method, the performance gaps are within $-2\%$ in all of the 11 entries, which includes 8 entries that *FeedSign* outperforms the baseline methods.

**Heterogeneity resilience.** Table 14 contains the results presented in Table 5, offering additional results for various settings of the Dirichlet distribution's control parameter $\alpha$ and training step $T$.

### H.5 WHY FINE-TUNING OVER FROM-THE-SCRATCH TRAINING

In language models, the approach uses prompts that ensure the objective is close to that of the pertaining in finetuning, guaranteeing its good performance. In image models, the most straightforward way to adapt a pre-trained model to a new dataset with different number of classes is to change the size of the classifier layer.

Table 12: Detailed results on RoBERTa-large over language tasks. Best results obtained using federated ZO optimization is **bolded**, and metric gap to that of FO method is reported in the rightmost column. We mark the performance gap between *FeedSign* and the best federated ZO method in a bracket.

| Task | SST-2 | SST-5 | SNLI | MNLI | RTE | TREC | Gap |
|------|-------|-------|------|------|-----|------|-----|
| Type | — sentiment — | | - natural language inference - | | | – topic – | |
| Zero-shot | 79.0 | 35.5 | 50.2 | 48.8 | 51.4 | 32.0 | – |
| $k = 16$ | | | | | | | |
| FO | 91.8 | 47.5 | 77.5 | 70.0 | 66.4 | 85.0 | – |
| MeZO | 90.5 | 45.5 | 68.5 | 58.7 | 64.0 | 76.9 | -5.6 |
| *ZO-FedSGD* | **89.7** | **46.8** | 63.1 | **60.5** | 63.1 | 70.0 | -7.5 |
| *ZO-FedAvg-1* | 89.3 | 46.5 | 68.5 | 59.9 | **66.0** | 73.8 | **-5.7** |
| *ZO-FedAvg-2* | 89.3 | 46.5 | 68.5 | 59.9 | **66.0** | 73.8 | **-5.7** |
| *FeedSign* | 88.9 | 45.0 | **69.7** | 59.7 | 65.3 | **75.6** | -5.8 |
| | (-0.8) | (-1.8) | (–) | (-0.8) | (-0.7) | (–) | |
| $k = 512$ | | | | | | | |
| FO | 93.9 | 55.9 | 88.7 | 84.4 | 82.7 | 97.3 | – |
| MeZO | 93.3 | 53.2 | 83.0 | 78.3 | 78.6 | 94.3 | -3.7 |
| *ZO-FedSGD* | 93.0 | 52.0 | **84.9** | 74.8 | 76.8 | 94.4 | -4.5 |
| *ZO-FedAvg-1* | 92.6 | 52.7 | 83.7 | 77.0 | **79.7** | 94.6 | -3.7 |
| *ZO-FedAvg-2* | **93.8** | **54.1** | 82.8 | **77.1** | 78.7 | **95.0** | **-3.5** |
| *FeedSign* | 92.6 | 50.4 | 83.1 | 76.0 | 74.3 | 93.0 | -5.5 |
| | (-1.2) | (-3.7) | (-1.8) | (-1.1) | (-5.4) | (-2.0) | |

Table 13: Detailed results on OPT-13B over language tasks. Best results obtained using federated ZO optimization is **bolded**, and metric gap to that of FO method is reported in the rightmost column. We mark the performance gap between *FeedSign* and the best federated ZO method in a bracket.

| Task | SST-2 | RTE | CB | BoolQ | WSC | WIC | MultiRC | COPA | ReCoRD | SQuAD | DROP | Gap |
|------|-------|-----|-----|-------|-----|-----|---------|------|--------|-------|------|-----|
| Type | | | — classification — | | | | | – multiple choice – | | — generation — | | |
| Zero-shot | 58.8 | 59.6 | 46.4 | 59.0 | 38.5 | 55.0 | 46.9 | 80.0 | 81.2 | 46.2 | 14.6 | – |
| FO | 92.0 | 70.8 | 83.9 | 77.1 | 63.5 | 70.1 | 71.1 | 79.0 | 74.1 | 84.9 | 31.3 | – |
| MeZO | 91.4 | 66.1 | 67.9 | 67.6 | 63.5 | 61.1 | 60.1 | 88.0 | 81.7 | 84.7 | 30.9 | -3.1 |
| *ZO-FedSGD* | 84.7 | 60.2 | 67.8 | 64.1 | 52.8 | 55.3 | 54.1 | 84.0 | **81.7** | 76.1 | 29.4 | -7.9 |
| *ZO-FedAvg-1* | 84.7 | 61.3 | 67.8 | **64.8** | 52.8 | 54.3 | 54.0 | 86.0 | 81.6 | 76.1 | 29.8 | -7.6 |
| *ZO-FedAvg-2* | 84.7 | **62.0** | **69.6** | 63.4 | 52.8 | 53.7 | 52.9 | 83.0 | 81.0 | 75.4 | **29.9** | -8.1 |
| *FeedSign* | **87.7** | **62.0** | 67.8 | 64.5 | **60.5** | **55.7** | **57.3** | **88.0** | **81.7** | **77.6** | 28.5 | **-6.0** |
| | (–) | (–) | (-1.8) | (-0.3) | (–) | (–) | (–) | (–) | (–) | (–) | (-1.4) | |

Table 14: More results on OPT-125M over language models with iid and non-iid data. We **bolded** the results that *FeedSign* performs equally with or better than *ZO-FedSGD*.

| Task | SST-2 | RTE | CB | BoolQ | WSC | WIC | MultiRC |
|---|---|---|---|---|---|---|---|
| Zero-shot | 51.2 | 53.0 | 48.2 | 41.5 | 37.5 | 51.2 | 49.7 |
| ——————— iid STEP=20000 ——————— | | | | | | | |
| *ZO-FedSGD* | 72.7 | 49.0 | 69.6 | 58.8 | 50.0 | 51.4 | 55.0 |
| *FeedSign* | **75.1** | **52.3** | 67.8 | **59.5** | **55.7** | **54.5** | 54.5 |
| ——————— non-iid STEP=20000 ——————— | | | | | | | |
| *ZO-FedSGD*, $\beta = 1.0$ | 58.4 | 48.0 | 50.0 | 41.8 | 36.5 | 51.4 | 45.0 |
| *FeedSign*, $\beta = 1.0$ | 55.0 | **48.0** | **58.9** | **43.1** | **36.5** | 50.6 | 44.6 |
| *ZO-FedSGD*, $\beta = 2.0$ | 79.8 | 52.7 | 58.9 | 62.6 | 63.4 | 49.5 | 55.5 |
| *FeedSign*, $\beta = 2.0$ | 79.2 | **52.7** | **62.5** | 62.1 | **63.4** | **50.7** | 54.6 |
| *ZO-FedSGD*, $\beta = 3.0$ | 78.7 | 53.0 | 62.5 | 62.8 | 63.4 | 50.1 | 55.5 |
| *FeedSign*, $\beta = 3.0$ | 78.6 | 52.3 | **64.2** | 61.8 | **63.4** | **51.2** | 54.6 |
| *ZO-FedSGD*, $\beta = 4.0$ | 81.0 | 52.3 | 62.5 | 62.6 | 61.5 | 50.0 | 55.5 |
| *FeedSign*, $\beta = 4.0$ | 79.7 | **53.4** | **67.8** | 61.4 | **63.4** | **51.4** | **55.5** |
| *ZO-FedSGD*, $\beta = 5.0$ | 81.1 | 53.0 | 41.0 | 61.8 | 61.5 | 50.4 | 55.4 |
| *FeedSign*, $\beta = 5.0$ | 79.4 | **53.4** | **46.4** | **61.9** | **63.4** | **50.4** | 55.1 |
| ——————— iid STEP=40000 ——————— | | | | | | | |
| *ZO-FedSGD* | 79.0 | 51.2 | 67.8 | 58.9 | 50.9 | 52.3 | 54.9 |
| *FeedSign* | **82.9** | **54.8** | **67.8** | **59.3** | **50.9** | **53.4** | **55.7** |
| ——————— non-iid STEP=40000 ——————— | | | | | | | |
| *ZO-FedSGD*, $\beta = 1.0$ | 64.1 | 47.6 | 57.1 | 41.9 | 36.5 | 51.0 | 44.7 |
| *FeedSign*, $\beta = 1.0$ | **66.9** | 47.2 | **67.8** | 41.5 | **36.5** | 50.9 | 44.5 |
| *ZO-FedSGD*, $\beta = 2.0$ | 82.3 | 52.7 | 64.2 | 62.6 | 63.4 | 50.1 | 55.5 |
| *FeedSign*, $\beta = 2.0$ | 82.1 | 52.3 | **66.0** | **63.1** | **63.4** | **50.3** | 55.4 |
| *ZO-FedSGD*, $\beta = 3.0$ | 81.1 | 52.7 | 66.0 | 62.7 | 63.4 | 50.3 | 55.5 |
| *FeedSign*, $\beta = 3.0$ | **82.9** | 52.3 | **66.0** | **62.7** | **63.4** | 50.1 | **55.6** |
| *ZO-FedSGD*, $\beta = 4.0$ | 82.2 | 53.0 | 64.2 | 62.7 | 65.3 | 50.1 | 55.5 |
| *FeedSign*, $\beta = 4.0$ | **83.0** | 52.7 | **67.8** | **62.7** | **65.3** | **52.3** | **55.5** |
| *ZO-FedSGD*, $\beta = 5.0$ | 82.1 | 52.7 | 58.9 | 62.6 | 65.3 | 51.0 | 55.5 |
| *FeedSign*, $\beta = 5.0$ | **82.9** | **52.7** | **66.0** | **65.3** | **65.3** | 50.0 | **55.5** |
| ——————— iid STEP=60000 ——————— | | | | | | | |
| *ZO-FedSGD* | 82.3 | 50.9 | 69.6 | 59.0 | 51.9 | 50.7 | 54.4 |
| *FeedSign* | **84.2** | **54.5** | 67.8 | **60.2** | 49.0 | **53.4** | **56.0** |
| ——————— non-iid STEP=60000 ——————— | | | | | | | |
| *ZO-FedSGD*, $\beta = 1.0$ | 70.7 | 47.2 | 64.2 | 40.6 | 36.5 | 50.3 | 44.6 |
| *FeedSign*, $\beta = 1.0$ | **73.0** | **47.2** | **66.0** | **40.8** | **36.5** | 50.0 | 44.5 |
| *ZO-FedSGD*, $\beta = 2.0$ | 81.1 | 52.7 | 64.2 | 63.0 | 63.4 | 50.1 | 55.5 |
| *FeedSign*, $\beta = 2.0$ | **82.5** | **52.7** | 62.5 | 62.6 | 62.5 | **50.1** | **55.5** |
| *ZO-FedSGD*, $\beta = 3.0$ | 81.5 | 53.0 | 66.0 | 62.8 | 63.4 | 50.6 | 55.5 |
| *FeedSign*, $\beta = 3.0$ | **82.9** | 52.7 | **66.0** | **63.0** | 61.5 | 50.4 | 55.2 |
| *ZO-FedSGD*, $\beta = 4.0$ | 83.2 | 52.3 | 66.0 | 62.8 | 63.4 | 49.5 | 55.5 |
| *FeedSign*, $\beta = 4.0$ | **83.3** | 51.9 | 62.5 | 62.6 | 59.6 | **51.4** | **55.6** |
| *ZO-FedSGD*, $\beta = 5.0$ | 81.7 | 52.7 | 62.5 | 62.4 | 63.4 | 50.4 | 55.5 |
| *FeedSign*, $\beta = 5.0$ | **83.4** | 51.9 | **62.5** | 62.1 | 59.6 | **50.6** | **55.7** |

