# OpenReview forum: "FeedSign: Full-parameter Federated Fine-tuning of Large Models with Extremely Low Communication Overhead of One Bit"
_ICLR.cc/2025/Conference — Submitted to ICLR 2025_

### Official Review · Reviewer_rcZ7 · 2024-11-01

**Soundness:** 3
**Presentation:** 2
**Contribution:** 2
**Rating:** 6
**Confidence:** 3

**Summary:**

This paper presents FeedSign, a federated fine-tuning (FFT) method that significantly reduces communication overhead for large models, requiring just one bit per step. Unlike conventional FFT, which demands high memory and communication, FeedSign uses zeroth-order optimization and a shared random generator to let clients signal only a binary vote indicating local model improvement. This keeps memory demands low and is compatible with a range of device constraints. FeedSign converges at rates similar to first-order methods and is robust against data heterogeneity and Byzantine attacks, performing on par with or better than other FFT methods across model sizes up to 13 billion parameters.

**Strengths:**

* Strong experimental results: the considered experimental setups are diverse and feature strong baselines, and the results of the proposed methods seem to be on par or better than the considered baselines.
* The idea of communication one bit of information is quite simple and could be used in practice.

**Weaknesses:**

* **Byzantine Resilience.** The paper could explore a broader range of Byzantine attacks and defenses, since “Byzantine resilience” refers to the ability of tolerating arbitrary corruptions. While robustness to basic Byzantine failures (Section 4.3) is a positive outcome, the method’s effectiveness against specific attacks (e.g., label flipping, gradient noise injection) is unclear. For instance, Allouah et al. (2023) explored representative defenses and attacks that might be relevant here, and integrating or testing such techniques could enhance FeedSign’s security profile.

* **Novelty Consideration.** Although FeedSign introduces an efficient mechanism for communication reduction, the reliance on sign-based updates and zeroth-order optimization is not entirely novel. Techniques like sign-SGD have been previously studied for Byzantine-resilient federated learning (Li et al. 2019), and zeroth-order methods are known in federated contexts. While FeedSign’s combination of these ideas is compelling, the novelty could be highlighted more by contrasting its specific contributions with these prior works in-depth.

* **Practical Relevance of Assumptions.** Assumption 2, in particular, appears to impose conditions that may not align with realistic settings. Further analysis or empirical evaluation showing how sensitive FeedSign’s performance is to deviations from this assumption would clarify the applicability of the theoretical results.

### References

Li et al. (AAAI 2019). RSA: Byzantine-robust stochastic aggregation methods for distributed learning from heterogeneous datasets.

Allouah et al. (AISTATS 2023). Fixing by Mixing: A Recipe for Optimal Byzantine ML under Heterogeneity.

**Questions:**

1. Could the authors clarify how FeedSign would handle more advanced Byzantine attack scenarios beyond "random" faults? Are there defenses that could be integrated into FeedSign to enhance robustness without substantially increasing communication costs?

2. The reliance on Assumption 2 for theoretical guarantees may limit FeedSign’s applicability in real-world data distributions. Could the authors provide empirical insights or adjustments to make this assumption more practical?

3. Since FeedSign’s communication relies on PRNG and zeroth-order methods, do these choices introduce any potential issues with reproducibility or variance in results across different clients, and if so, how might they be mitigated?

---

> ### Author Response · Authors · 2024-11-19
> **Response to Reviewer rcZ7**
>
> **W1: Byzantine resilience**.
>
> **A1**: Thanks for the insightful comment. In response to your comment, we have revised Remark 4 (lines 340-346) to discuss why other attack methods are excluded from our investigation.
>
> - FeedSign is immune to gradient noise injection, which damages convergence by altering the gradient direction. This is because in FeedSign-like algorithms (uses SPSA, Def. 1), the gradient is represented by $\boldsymbol{g}=p \boldsymbol{z}$, where $p$ is the gradient projection (a scalar), and $\boldsymbol{z}$ is the PRNG generated Gaussian random vector. The deterministic nature of PRNG guarantees that $\boldsymbol{z}$ will be identical as long as the same PRNG is used, so the only way to disrupt $\boldsymbol{g}$ will be changing $p$, which is the demonstrated case in our experiment section.
>
> - The same reason applies to label flipping, and every attack means against gradient estimation boils down to an inaccurate measurement of $p$.
>
>
> To wrap up, our chosen baseline ZO-FedSGD could have complex designs regarding projection modification, but in FeedSign, Byzantine clients have only one bit to manipulate.
>
> **W2: Novelty Consideration**.
>
> **A2**: Thanks for raising this issue. We would like to point out that "1 bit" in our context fundamentally differs from "1 bit" in signSGD-like works. In signSGD, "1 bit" is elementwise; that is, each entry of the gradient vector, rather than the whole gradient being quantized to "1 bit". Consequently, utilizing signSGD-like algorithms will result in a dimension-dependent communication overhead, while our "1 bit" is dimension-independent. We added discussions to our paper in light of this comment (lines 091-093, 163-166, and 173-174).
>
> **W3: Practical Relevance of Assumptions**.
>
> **A3**: Thanks for the insightful comment. Assumption 2 is a less common assumption on similar optimization problems, as pointed out. Assumption 2 is saying that most of the eigenvalues of the Hessian matrix of the objective should be near 0. Intuitively, if $r$ is large, then there will be many eigenvalues ($\geq r$) having magnitudes comparable to the operator norm. Many works empirically show this is true in a well-trained model. In light of this, we added references to earlier works that empirically verified this property (lines 317-318).
>
> **Q1: More advanced Byzantine attacks**.
>
> **A4**: Thanks for the question. We would like to point out that due to the reason shown in **A1**, since the direction of the gradients ($\boldsymbol{z}$ part) is fixed due to the deterministic nature of PRNG, to the best of our knowledge, every attack means will boil down on having an inaccurate $p$, which is the setting we adopted in experiments.
>
> **Q2: Reliance on local low effective rank assumption**.
>
> **A5**: Thanks for the question. We would like to clarify that reliance on Assumption 2 of this work is the reason we restrict our domain of discussion to FFT rather than general FL. To the best of our knowledge, there is no adjustment that will result in a considerably better analysis or empirical results. However, as discussed in **A3**, Assumption 2 is acceptable and is supported by empirical results.
>
> **Q3: Choices of PRNG**.
>
> **A6**: Thanks for the question. As long as a high-quality PRNG is chosen (done by computing device manufacturers like NVIDIA) and a zeroth-order optimizer that does not introduce further randomness is chosen, reproducibility is mathematically guaranteed. Client devices only need to worry about agreeing to use the same PRNG in the same way.

---

> > ### Comment · Reviewer_rcZ7 · 2024-11-25
> >
> > I thank the authors for their rebuttal and would like to maintain my original score.

---

### Official Review · Reviewer_G6oG · 2024-11-02

**Soundness:** 3
**Presentation:** 2
**Contribution:** 2
**Rating:** 5
**Confidence:** 4

**Summary:**

This paper introduces FeedSign, a federated fine-tuning framework that employs zeroth-order (ZO) optimizers and pseudo-random number generators, aiming to reduce communication overhead from 64 bits to 1 bit per communication round. While this technical innovation represents a noteworthy advancement in communication per round, the practical relevance of such a reduction remains debatable, particularly given the existing sufficiency of bandwidth for 64 bits. Furthermore, FeedSign does not demonstrate a clear performance advantage over existing frameworks such as FwdLLM and FedKSeed, which undermines its overall contribution.

**Strengths:**

1 The motivation of this paper is well shown.
2 Detailed proof is given.

**Weaknesses:**

1 Lack of convergence speed analysis. The paper lacks a comprehensive analysis of the convergence speed, which is critical as it seems to be the tradeoff between FeedSign and conventional methods (like FedAvg). Adding a direct comparison can better validate the proposed method.

2 Performance Comparison Deficits: Despite its lower communication cost, FeedSign does not surpass ZO-FedSGD in several performance metrics across different experiments. The marginal reduction of 63 bits in communication cost per round is not adequately justified as a significant advantage over the existing methods, especially when FeedSign does not exhibit faster convergence. This could result in an increased number of communication rounds, negating its low bandwidth benefits. Figure 2 underscores this issue, showing that ZO-FedSGD can achieve faster convergence speed.

3 Vague Claims of Superiority: The paper claims that FeedSign offers “numerous surprising benefits” over its predecessors. However, these benefits are not well-defined or substantiated beyond the reduced communication requirement, making these claims appear overstated. The advantages over the frameworks in [1], [2], and [3] are ambiguously presented and lack clear support. The contribution should be strengthened.

4 Limited Client Participation: The experimental setup with only five participating clients raises questions about the scalability and generalizability of FeedSign. Testing with a larger client pool could provide more robust insights into the framework’s performance across diverse federated environments.

[1] Mengwei Xu, Dongqi Cai, Yaozong Wu, Xiang Li, and Shangguang Wang. {FwdLLM}: Efficient federated finetuning of large language models with perturbed inferences. In 2024 USENIX Annual Technical Conference (USENIX ATC 24), pp. 579–596, 2024
[2] Zhen Qin, Daoyuan Chen, Bingchen Qian, Bolin Ding, Yaliang Li, and Shuiguang Deng. Federated full-parameter tuning of billion-sized language models with communication cost under 18 kilobytes.
[3] Sadhika Malladi, Tianyu Gao, Eshaan Nichani, Alex Damian, Jason D Lee, Danqi Chen, and Sanjeev Arora. Fine-tuning language models with just forward passes.

**Questions:**

See Weakness.

---

> ### Author Response · Authors · 2024-11-19
> **Response to Reviewer G6oG**
>
> **W1: Lack of convergence speed analysis**
>
> **A1**: Thanks for the comment. In response to this comment, we have newly included a convergence analysis of the proposed method and the conventional FO-based method (lines 302-304) in Theorem 1, allowing a direct comparison of the convergence speed among the considered methods (lines 315-316).
>
> **W2: Performance comparison deficits**
>
> **A2**: Thanks for the comment. It is true but natural that the lower communication overhead of FeedSign results in slower convergence. However, FeedSign enjoys the following advantages compared to ZO-FedSGD:
>
> - Data heterogeneity resilience. In rationale, FeedSign trades for better resilience against data heterogeneity at the cost of having a slightly lower convergence speed, as shown in Theorem 1 and Remark 3. Convergence analysis of FeedSign yields a performance bound that is independent of data heterogeneity. This is further verified empirically in Figure 2 and Table 5. Notably, FeedSign outperforms ZO-FedSGD in the practical noniid case.
>
> - Byzantine resilience. When Byzantine clients are in the client pool, the performance of ZO-FedSGD is compromised, while FeedSign shows marginal performance loss. This is shown in Tables 4, 6, and 7, as well as the newly appended Figure 3, in response to this comment. Additionally, the one-bit design also closes the room for further Byzantine attack design since the sign is the only thing that can be manipulated. This is different from ZO-FedSGD, where Byzantine clients can choose a cleverly distorted distribution of gradient projection to disable the system more effectively or stealthily. We have highlighted this point in Remark 4.
>
>
> **W3: Vague Claims of Superiority**
>
> **A3**: Thanks for the comment. We revised the corresponding part by further strengthening our claim with mathematical and empirical evidence.
>
> - As recalled in our response to W2, we mathematically characterize data heterogeneity resilience by showing that FeedSign's convergence speed and error bound are irrelevant to data heterogeneity factors. We highlighted this claim in Remark 3 in response to this comment.
> - We also characterize how the Byzantine client affects FeedSign's convergence, as highlighted in Remark 4. We quantify how the convergence is compromised by two key factors, the Byzantine client proportion and the inherent sign reversing probability, combining results in Theorem 1 and Proposition 1.
> - The mathematical findings are backed by empirical results. We demonstrate that FeedSign can outperform ZO-FedSGD under a noniid case in Figure 2, and we include Figure 3, reporting the loss and accuracy curve versus the number of steps elapsed. FeedSign manifests good Byzantine resilience with marginal performance degradation, while ZO-FedSGD is compromised by the Byzantine client.
>
> Moreover, we discuss some underexplored features FeedSign-like algorithms bring to an FL system, including parameter security and efficient model sharing.
>
> **W4: Limited Client Participation**
>
> **A4**: Thanks for the comment. In response to your comment, we will be adding a comparison of OPT-125M with client pool sizes K=25 in the experiment section as soon as the results are available.

---

> ### Comment · Reviewer_G6oG · 2024-11-26
>
> Thank you for the response. However, I still find the contribution of the proposed method unclear. While the newly added results and analyses provide additional insights, they do not convincingly demonstrate that the proposed method offers a distinct advantage in handling data heterogeneity. For instance, in Table 5, ZO-FedSGD outperforms the proposed method in 4 out of 7 non-IID tasks, raising questions about the claimed resilience.
>
> If Byzantine resilience is indeed the main strength of FeedSign, the current experiment setup appears overly simplistic. A more rigorous evaluation, such as considering varying rates of Byzantine clients, would provide stronger evidence of robustness.
>
> Additionally, the tradeoff in convergence speed does not present a compelling advantage. The improvements seem marginal and are not adequately justified as a significant contribution.
>
> Regarding the high-level features discussed in Section 5, it seems that their advantages stem primarily from seed-projection, which is not exclusive to FeedSign and, therefore, cannot be claimed as a direct contribution of the method. Moreover, the analysis accompanying Table 4 is somewhat unclear. While ZO-FedSGD demonstrates superior performance in the table, the authors conclude that FeedSign is faster than the ZO-based training SOTA, which appears contradictory.
>
> Lastly, in Figure 3, the distributions for the upper and lower subfigures are missing.

---

> ### Author Response · Authors · 2024-12-01
> **Response to Follow-Up Comment by Reviewer G6oG**
>
> Thanks for your follow-up comments.
>
> **Empirical support for implied data heterogeneity.** As pointed out, ZO-FedSGD outperforms our method in 4 out of 7 instances. However, it should be noted that in these instances, the gap is smaller (-0.3 in WIC, -0.1 in MultiRC), including two draws (0.0 in RTE and WSC). In those instances where we outperform ZO-FedSGD, we observe bigger leads (+2.3 in SST-2, +1.8 in CB, +0.2 in BoolQ). We report the performance with a larger client pool size.
>
> $\beta=1.0, K=25$,
>
> |     | SST-2 | RTE | CB  | BoolQ | WSC | WIC | MultiRC |
> | --- | --- | --- | --- | --- | --- | --- | --- |
> | ZO-FedSGD | 58.8 | 52.7 | 60.7 | 52.6 | 42.3 | 50.3 | **55.8** |
> | FeedSign | **64.2** | **53.7** | **66.0** | **58.4** | **44.2** | **51.2** | **55.8** |
>
> It can be observed that FeedSign outperforms ZO-FedSGD in most of the columns, reflecting the theoretically implied data heterogeneity resilience.
>
> **Simple settings in Byzantine resilience experiments.** Thanks for your comment. We report the test accuracy of ZO-FedSGD and FeedSign with different rates of Byzantine clients as follows with the number of steps aligned. Head of the table $K_B/K$ denotes $K_B$ Byzantine clients in overall $K$ clients.
>
> | CIFAR-100 | 1/25 | 2/25 | 3/25 |
> | --- | --- | --- | --- |
> | ZO-FedSGD | 8.1 | 4.0 | 3.2 |
> | FeedSign | **25.7** | **19.6** | **5.2** |
>
> | CIFAR-10 | 1/25 | 2/25 | 3/25 |
> | --- | --- | --- | --- |
> | ZO-FedSGD | 82.5 | 71.0 | 54.0 |
> | FeedSign | **92.4** | **90.9** | **82.1** |
>
> We would like to mention that under this harsher setting, all of the CIFAR-100 and some of the CIFAR-10 instances do not reach convergence at the same number of steps as that used in the paper. However, FeedSign has shown its advantage in convergence speed compared to ZO-FedSGD under these settings.
>
> **Convergence speed tradeoff.** Thanks for the comment. If our understanding is correct, the reviewer thinks that the trade for data heterogeneity is not good enough. We would like to point out that apart from data heterogeneity resilience, reflected both theoretically and empirically, FeedSign also has good resilience against Byzantine attacks, with a significantly lower communication overhead. We believe that this is a valid advantage and can be of good value under certain circumstances.
>
> **Regarding section 5.** Thanks for your comment. It is true that the advantages discussed in Section 5 stem from the seed-projection design. We have carefully phrased our statement in our previous submission as
>
> - "discuss some interesting advantages as byproducts guaranteed by the design of FeedSign" in the Abstract,
>
> - "discuss some interesting features as byproducts" in the Introduction,
>
> - "in FL systems featuring alike designs to FeedSign", "FeedSign-like seed-projection pairs design" in Section 5,
>
>
> and made no claims that this is exclusive to FeedSign or is one of our main contributions. We added this discussion to our work because we believe they can be interesting and valuable features but remain underexplored in recent pioneering works.
>
> **Regarding Table 4.** We refer *ZO-based training SOTA* to methods that start ZO optimization from a randomly initialized network (from scratch), and both ZO-FedSGD and FeedSign should be classified into *ZO-based finetuning*. Our statement is that given FeedSign is an aggressive scheme that discards most of the knowledge on client gradient compared to ZO-FedSGD, it can still outperform ZO-based training SOTA in a relatively small number of steps thanks to a general pre-trained model, as shown in the following table, extracted from our previous submission.
>
> | Dataset | CIFAR-10 | CIFAR-100 |
> | --- | --- | --- |
> | ZO-trained SOTA (from scratch) | 86.5 | 34.2 |
> | FeedSign | **91.7** | **45.3** |
>
> We believe that this finding gives us an empirical sense that finetuning should be considered a priority if we want to use ZO optimizers on NN tasks.
>
> **Regarding Figure 3.** If we understand correctly, the reviewer is referring to the confidence interval that appears as shades around the curves. The confidence interval is actually displayed in Figure 3. We report the standard deviation of metrics in the last epoch in 5 repeats as follows.
>
> |     | Loss | Accuracy |
> | --- | --- | --- |
> | CIFAR-10 | 0.07 | 0.10 |
> | CIFAR-100 | 0.05 | 0.19 |
>
> It took us some time to prepare evidence to support our response. However, we will respond ASAP from now on, as the discussion phase is closing. We greatly appreciate your time and attention in reviewing our paper.

---

### Official Review · Reviewer_WF4n · 2024-11-03

**Soundness:** 1
**Presentation:** 2
**Contribution:** 3
**Rating:** 5
**Confidence:** 4

**Summary:**

The paper proposes FeedSign and ZO-FedSGD as two methods to fine tune large models in an FL setting with communication efficiency. They utilize zeroth-order gradient estimation to also reduce the computational complexity in the edge devices. In the ZO-FedSGD method, each client samples a seed and computes the gradient estimator coefficient and shares the seed and the coefficient with the server (64 bits per round). In the FeedSign method, the seed is shared by the server to all clients, and clients compute their gradient estimator coefficient for that seed and send the sign of that estimator to the server (1 bit per round). In Theorem 1, they proved that the method converges with exponential rate O(e^{-t}). They evaluate the method's performance under different models and datasets for different tasks. Also the differential private version of FeedSign (DP-FeedSign) is proposed in theorem 2.

**Strengths:**

The paper introduces a novel approach to minimizing communication overhead in fine-tuning large models by communicating only one bit per round, regardless of model size, structure, or task. This method combines communication efficiency with the computational efficiency of a zeroth-order gradient estimator in each client, which reduces memory requirements by estimating gradients through inference rather than backpropagation. The methods for both ZO-FedSGD and FeedSign are detailed in Algorithm 1. For the FeedSign method, which employs a voting mechanism, robustness against Byzantine attacks is achieved.
The exponential convergence rate O(e^{-t}) is proved in Theorem 1, and a comprehensive set of experiments is provided for various model architectures (ResNet, ViT, RoBERTa, OPT) and scales (ranging from 11M to 13B parameters).
The paper addresses the critical challenge of communication in federated learning by reducing communication overhead to one bit per round for FeedSign and 64 bits for ZO-FedSGD. It also tackles computational limitations in edge devices through the use of a zeroth-order gradient estimator. The method scales effectively to very large models (up to 13B parameters) and offers additional benefits, such as enhanced parameter security and privacy guarantees as byproducts.

**Weaknesses:**

1- The ZO-FedSGD method is a distributed version of a zeroth-order algorithm, similar to MeZO, and is referenced in other forms in some citations within the paper. However, it does not appear to be the main contribution of the paper; instead, FeedSign is presented as the primary contribution, which is theoretically and practically distinct from previous work.

2- The assumption that FeedSign is unbiased is fundamentally inaccurate. In FeedSign, only the sign of $p$ is transmitted and aggregated, which would only yield unbiasedness if all $p$ values were uniformly $+1$ or $-1$. Since this is not the case, Lemma 1 holds only for ZO-FedSGD and does not apply to FeedSign. Consequently, it cannot be used in Theorem 1, and the proof of convergence for FeedSign in Theorem 1 should be revised.

3- In the experimental section, Table 1 shows instances where FeedSign or ZO-FedSGD achieve better accuracy than the centralized version, which seems improbable under the same scenario, as a decentralized version typically cannot surpass the accuracy of its centralized counterpart. If different scenarios were used for fine-tuning, this should be specified in the section, along with a rationale for selecting particular scenarios for the experiments.

**Questions:**

1- Your method clearly introduces bias in gradient estimation by using only signs. Have you analyzed how this bias affects the convergence compared to unbiased ZO-FedSGD? Why does FeedSign still work well despite being biased?
2- Is there any theoretical bound on the bias of the method?
3- Is there any theoretical prove for the convergence of the FeedSign considering its a biased estimation of the gradient?

---

> ### Author Response · Authors · 2024-11-19
> **Response to Reviewer WF4n (Part 1)**
>
> **W1: Vague primary contribution**
>
> **A1**: Thanks for the comment. We would like to admit that the primary idea of zeroth-order finetuning of large models was originally explored by MeZO and ZO-FedSGD. We make further advances from the prior works in the following perspectives.
>
> - Compared to ZO-FedSGD, FeedSign is more resilient to data heterogeneity and Byzantine attacks and has an order-of-magnitude smaller communication overhead due to the more restrained gradient projection representation of 1 bit. The conclusions are backed by theoretical and empirical results.
>
> - We further discovered some underexplored features of ZO optimization on FL systems, including parameter security, efficient model sharing, and its potential differential privacy enhancement.
>
> - We also extend the investigation to vision models, including ViT and ResNet, and demonstrate the advantage of ZO finetuning compared to from-the-scratch ZO training (lines 401-409).
>
>
> **W2: Convergence analysis issue**
>
> **A2**: Prompted by your comment, we carefully checked our proof and found that the convergence analysis of FeedSign does not need the unbiased gradient estimate assumption. We are sorry that we have mistakenly stated that the convergence analysis of FeedSign is based on the assumption of unbiasedness in the previous version. We have revised Theorem 1 and rewritten the proofs to fix this issue (see lines 936-989 in the updated version). The new proof is sketched as follows.
>
> We start from the L-smooth assumption.
>
> Two key steps to arrive at the descent lemma for FeedSign are to calculate the inner product term and the quadratic term.
>
> For the inner product term, the weight distance $\mathbf{w}_{t+1} - \mathbf{w}_t$ term will be replaced with an estimated gradient projection-related term, and Assumption 5 further replaces it with a true gradient-related term. We show that this term is eventually half-normal distributed, whose expectation can be easily derived.
>
> For the quadratic norm term, since FeedSign records no amplitude, the norm of the weight difference $\|\boldsymbol{w}_{t+1} - \boldsymbol{w}_t\|_2^2$ is fixed to a constant related to the Lipschitz constant $L$ and learning rate $\eta$. We will reach Equation 43 after this.
>
>
> **W3: Abnormal outperforming instances**
>
> **A3**: Thanks for pointing out this issue. We have elaborated on the specificities of the experiment settings in the corresponding part (lines 361-363). To be brief, we copied the reported results in the MeZO paper for the FO and MeZO rows for fair comparison. In runs of ZO-FedSGD and FeedSign, we keep the total number of forward passes (dominant part of computational cost, $n_{\text{centrl}} = T$) the same as that in MeZO. However, different from centralized methods, the number of forward passes does not equal to steps, specifically, reciprocal smaller regarding the client pool size ($n_{\text{distrb}}=n_{\text{centrl}} / K=T$). Hence, it is equivalent to ZO-FedSGD and FeedSign adopting an early stop strategy, and centralized methods seem to have experienced overfitting in the referred instances.
>
> However, we realized that this could lead to misunderstandings, so we are running new results with a total number of *global model update steps* rather than a total number of *forward passes* aligned.

---

> ### Author Response · Authors · 2024-11-19
> **Response to Reviewer WF4n (Part 2)**
>
> **Q1: Bias effect**
>
> **A4**: Thanks for the question.
>
> Yes, we have added the analysis of the impact of biased gradient estimation in the updated version in response to this question. The consequence of biased gradient estimation is that FeedSign converges to the neighborhood of the stationary point with a fixed loss floor regardless of the data heterogeneity factors. In comparison, the error floor vanishes for the unbiased gradient estimation-based ZO-FedSGD under an ideal iid case ($c_g = 1, c_h = \sigma_h = 0$ in Assumption 3) but scales to the data heterogeneity factors. In plain words, the proposed design trades for better resilience against data heterogeneity at the cost of having a small constant error floor.
>
> As for why FeedSign works well with biased gradient estimates, if we keep the learning rate $\eta$ small, since the projected error floor scales to $\eta$, the error floor bound can be controlled within a small scale. This is reflected in the experiments and justifies the comparable performance of FeedSign.
>
> **Q2: Theoretical bound on biasedness**
>
> **A5**: Thanks for the question. In response to this question, we have added a subsection in the appendix (lines 1060-1079) discussing a special case where FeedSign is an unbiased estimator. Briefly, we show that when the true gradient projection $p$ takes value in $[-1, 1]$ and the difference between the batch gradient projection estimate $p_1$ and $p$, denoted by $n$, follows a uniform distribution on $[-1, 1]$, FeedSign is unbiased. We believe that the bias can be bounded by the divergence between the abovementioned distribution and the actual one of $n$. We are leaving this for future work as this may involve substantial workloads deserving a separate work.
>
> **Q3: Theoretical characterization of FeedSign**
>
> **A6**: Thanks for the question. As recalled in our reply to W2, we have revised the theorem and the proofs. We believe that the updated Theorem 1 is valid for the convergence analysis for FeedSign. If you should have further problems, please don't hesitate to reach out to us.

---

> > ### Comment · Reviewer_WF4n · 2024-11-26
> > **Revised Theorem Proof and Simulation Weaknesses**
> >
> > Thank you to the authors for their revisions and for updating the proof of the theorem. The revised proof appears convincing; however, the weaknesses in the simulations remain a concern, leading me to maintain my previous decision.

---

### Official Review · Reviewer_Qg9z · 2024-11-04

**Soundness:** 3
**Presentation:** 3
**Contribution:** 1
**Rating:** 5
**Confidence:** 4

**Summary:**

This work studies Federated fine-tuning (FFT) realized by utilizing zeroth-order (ZO) optimizers. By using shared pseudo-random number generators (PRNG) across devices, the authors suggest to compress the transmitted gradients using 1-bit; which is in turn robust, as a result of the associated low-influence per-client. A convergence analysis is provided, demonstrating exponential rate similarly to the first-order (FO) methods. An experimental stay and a discussion are further presented, where the latter extends the proposed algorithm into a differentially private one.

**Strengths:**

Quality & Clarity:
- The paper is well written and presented.
- The provided convergence analysis supports the validation of the presented algorithm.
- The discussion section is comprehensive.
- The reference to ZO-FedSGD is clear and easy- to-follow.

**Weaknesses:**

Originality & Significance:
- Compressed FL has been extensively studied in literature. Specifically, the idea of clients voting via 1-bit compression and shared source-of-randomness; its differentially private version; and its implied associated robustness due to the implicit low-influence of its clients; have all already been presented in: 1) Lang, Natalie, et al. "CPA: Compressed private aggregation for scalable federated learning over massive networks." ICASSP 2023-2023 IEEE International Conference on Acoustics, Speech and Signal Processing (ICASSP). IEEE, 2023, and in 2) Lang, Natalie, et al. "Compressed private aggregation for scalable and robust federated learning over massive networks." arXiv preprint arXiv:2308.00540 (2023).‏
- The setting of both works 1) and 2) is the conventional FL with FO optimization, rather than the less adopted FFT and ZO, respectively.
- Moreover, as reflect from Algorithm 1 and lines 46-47; FFT does not change the implementation and associated challenges of regular FL, having the latter not-restricted to models of a certain size.

**Questions:**

1. In lines 85-87: "However, we show that the per-step uplink and downlink communication overhead can be further reduced to 1 bit per step regardless of model size with little performance loss but numerous surprising benefits...". According to you Algorithm 1, only your uplink communication is being compressed. Additionally, in "numerous surprising benefits" the use of "numerous" may be too exaggerating.
2. In lines 187-188: "However, to the best of our knowledge, efforts toward communication efficiency, data heterogeneity, and Byzantine resilience on FL are separated, motivating this work.". How does you method supports data heterogeneity?
3. In line 227: "...perturb its model in the same direction." What do you mean by that?
4. In line: "FeedSign left the sampling of random seeds to the PS...". Is this left of sampling is unique to FeedSign or can also be implemented in ZO-FedSGD?
5. What is 'D' in eq. (16)?
6. In lines 315-318: "...the only means of damaging convergence of FFT due to the binary voting scheme in FeedSign is to always send a reversed sign to PS." Does sending a constant sign (regardless of the true value) is also a possibility?

---

> ### Author Response · Authors · 2024-11-19
> **Response to Reviewer Qg9z (Part 1)**
>
> **W: Originality and Significance**
>
> **A1**: Thanks for bringing out this insightful point. We would like to clarify the difference between FeedSign and the referred work in the following aspects:
>
> - In principle, FL's long-existing bottleneck lies in communicating model updates. Most efforts addressing this issue are separately doing **accurate** gradient estimation followed by **lossy** compression, leading to potentially unnecessary computational loads, as the compression eventually negates the costly effort of acquiring an accurate gradient estimation by backpropagation. In light of this, we envisage a more integrated and efficient framework that runs on gradient estimation that is less accurate but attainable and communicable with lower overheads with marginal performance loss, arriving at the proposal of FeedSign. This marks a fundamental difference between our work and conventional compression FL methods.
> - In consequence, FeedSign possesses a key advantage over FO-based methods with its largely reduced memory and computational load, which is critical in large model FFT. The significant memory overhead reduction (1/100 - 1/10, depending on model architectures and tasks) comes from FeedSign's exemption of backward passes due to the adoption of ZO optimization. The computational cost will also be largely cut down since FeedSign excludes backpropagation to perform gradient estimation, which is computation-intensive (especially for large models) but inevitable in FO-based methods.
> - Additionally, we introduce a new layer of privacy in FL systems. In [1,2], the server must estimate and host the global model for broadcast. However, in FeedSign, the server only collects and forwards seed-projection pairs submitted by clients. As a result, **not only data but also models are kept local and private** in FL systems that feature an aggregation method like FeedSign, adding double protection of privacy. Also, the hardware demand for the server is significantly lowered.
>
> Due to the abovementioned features, we believe that FeedSign possesses originality and significance. To highlight the originality and significance of our work, especially the differences from the referred work, we have included more discussions on the differences to the referred work regarding the commented issues (see lines 174-185, 493-495).
>
> **Q1: Rephrasing overstated statements**
>
> **A1**: Thanks for your question. We originally used the "numerous surprising benefits" to refer to the data heterogeneity, Byzantine resilience, parameter security, and other features brought by FeedSign while reducing the communication, computation, and memory overhead. We have rephrased the corresponding part of the paper and specified the benefits (see lines 087-089).
>
> **Q2: Vague support of data heterogeneity resilience**
>
> **A2**: Thanks for the comment. FeedSign's data heterogeneity resilience property is demonstrated both theoretically (see Remark 3, lines 319-323) and empirically (see Figure 2 & Table 5). Theoretically, our analysis shows that FeedSign trades we trade for more resilience against data heterogeneity at the cost of having a fixed error floor. Empirically, FeedSign shows a smaller performance gap than the baseline under data heterogeneity. We have refined the corresponding part in response to this question.
>
> **Q3: Regarding the definition of perturbation**
>
> **A3**: Thanks for the question. As shown in Def 1 (lines 216-223), the gradient estimation in FeedSign (and other algorithms using SPSA variants) is realized by clients measuring the induced change of the loss value after making a deliberate change on its model weights. This change is usually numerically small, hence referred to as *perturbation* in earlier works.
>
> **Q4: Seed sampling**
>
> **A4**: Thanks for the question. This question is related to a detailed design of FeedSign to realize its strengthened ability. In ZO-FedSGD, each client samples different seeds to perform their own gradient estimation; that is to say, they measure the change of loss towards different directions in the parameter space and update consistently according to seed-projection pairs reported by other clients forwarded by the server. In FeedSign, the projection is compressed to only one bit. Since allowing the clients to explore different directions while using only a binary projection could introduce too much noise, we decided to have them explore the same direction, sampled by the server. Clients upload their binary projections in the same exploring direction, like voting, which is a model usually used in FL compression or security research, as commented. Having a shared seed is also doable in ZO-FedSGD, but it could undermine the performance of the baseline since fewer update directions are explored. However, we did not dig deep into this due to the length limit.

---

> > ### Comment · Reviewer_Qg9z · 2024-11-20
> >
> > Hi, thanks for you answer. Following that:
> > - Your are saying that you introduce a new layer of privacy in FL systems as the server only collects and forwards seed-projection pairs submitted by clients. But in you paper, you said that the works of FwdLLM Xu et al. (2024) and FedKSeed Qin et al. (2023) discuss a federated fine-tuning framework that exchanges models by exchanging seed-projection pairs (lines 150-153). So, is that privacy contribution, relying on exchanging seed-projection, was already introduced by FwdLLM and FedKSeed?
> > - When pointing out the differences between the works 1) and 2) and yours, you say that the computational cost will be largely cut down since FeedSign excludes backpropagation to perform gradient estimation, which is computation-intensive (especially for large models) but inevitable in FO-based methods. FwdLLM and FedKSeed also exclude backpropagation as they utilize ZO optimization. So do they similarly enjoy this reduced computational cost?
> > - In light of these two trends, privacy and computational cost reduction have already been presented in the algorithms of FwdLLM and FedKSeed?
> > - As pointed out by you, the global model is not stored at the server, and is therefore not being communicated from the users to the server in the uplink channel; what is usually considered as the main motivation for utilizing compression in FL. So in your case, is the gain in your compressed suggestion is the memory overhead reduction in the local storage footprint at the remote edge users devices? (with the additional follow-up contributions of having each user characterized with low influence of one-bit)

---

> > > ### Author Response · Authors · 2024-11-20
> > > **Response to follow-up comments of Reviewer Qg9z**
> > >
> > > Hi, thanks for your follow-up review. In response:
> > >
> > > - Yes, regarding this additional privacy layer, this property is also valid in *FwdLLM* and *FedKSeed*. However, this benefit remains underexplored in their works. Additionally, from a broader view, *FeedSign* is more private than *FwdLLM* and *FedKSeed*, since it releases less information about the gradient projection estimates, apart from other resilience enhancements. We will clarify this in the next version of our paper in response to this comment.
> > >
> > > - Yes, they also enjoy this reduction in computational cost since we all employ ZO optimizers.
> > >
> > > - We would like to clarify that our work shares the benefit of computational cost reduction compared to the referred work in your earliest comment [1,2] as a result of employing ZO optimization. We discussed the additional "models kept local" privacy offered by the framework design since it is underexplored, and we consider this property valuable. Additionally, we would like to highlight the enhanced resilience of *FeedSign* against data heterogeneity and Byzantine attacks as a consequence of the one-bit compression while maintaining low memory, computation, and communication overheads for the clients.
> > >
> > > - We would like to clarify that uplink communication load reduction is a part of the motivation for the one-bit compression we employed. The feature that the server can host no actual global model is an attainable advantage of using ZO optimizers in a distributed manner. It is true that our compressed suggestion results in a reduction of memory overhead and storage footprint, but the benefit lies more in the resilience brought by the low influence and vague update.
> > >
> > >
> > > Thanks for your comment. If you should have further comments, please don't hesitate to reach out to us.
> > >
> > > [1] Lang, Natalie, et al. "CPA: Compressed private aggregation for scalable federated learning over massive networks." ICASSP 2023-2023 IEEE International Conference on Acoustics, Speech and Signal Processing (ICASSP). IEEE, 2023.
> > >
> > > [2] Lang, Natalie, et al. "Compressed private aggregation for scalable and robust federated learning over massive networks." arXiv preprint arXiv:2308.00540 (2023).

---

> ### Author Response · Authors · 2024-11-19
> **Response to Reviewer Qg9z (Part 2)**
>
> **Q5: Regarding the meaning of a parameter**
>
> **A5**: Thanks for the question. $D$ is a parameter introduced during applying the mean value theorem on a term in the proof ($ 2\sqrt{ab} \leq Da + b/D $). We have gone through our proof and discarded this parameter for a more precise result (see lines 294-311).
>
> **Q6: Regarding the attack policy**
>
> **A6**: Thanks for the question. The mentioned method is viable, and as pointed out, "only" is misleading. We emphasized the "always reverse" policy because we found that according to the analysis, the convergence of FeedSign is connected to how often the reported sign is "wrong". So, always sending a reversed sign should be a more effective attack policy. We have corrected that sentence (lines 322-323) in response to this question.

---

> ### Comment · Reviewer_Qg9z · 2024-11-26
>
> Thank you for your detailed response. While I appreciate your clarifications and proposed revisions, my concern regarding the limited novelty remains, so I will maintain my original score.

---

### Comment · Area_Chair_B5dx · 2024-11-26
**Response**

Dear Reviewers,

The authors have provided their rebuttal to your questions/comments. It will be very helpful if you can take a look at their responses and provide any further comments/updated review, if you have not already done so.

Thanks!

---

### Author Response · Authors · 2024-12-02
**Rebuttal Summary (part 1)**

We would like to thank the AC and PCs for handling our paper and the reviewers for their expertise, effort, and time in reviewing our submission. They have helped us a lot in optimizing our work. Since the discussion phase is closing, we summarize our claimed contributions and the rebuttal as follows.

**Claimed contributions.**

- We proposed FeedSign, a federated finetuning framework. It features a zeroth-order optimizer, splits the gradient estimation into 1) direction, and 2) projection, and asks the client to communicate only the sign of the gradient projection. Since 1) can be easily reconstructed in a remote device with a shared PRNG, the sign can be regarded as a vague update that uses up only 1 bit per step per client, independent of model size. Meanwhile, the vague updates boost the system with

  - data heterogeneity resilience,

  - and Byzantine attack resilience.


  The convergence of the FL system with the sign-only transmission scheme is theoretically characterized and empirically verified. FeedSign shares low memory and computation requirements since employing ZO optimizers while the per-step communication load is further reduced. We also discussed some underexplored but valuable features that can be found in similar seed-projection designs.


**Acknowledged strengths.**

- **Strong experiment settings.** Strong baselines are considered, and the results cover a wide range of models and tasks. `rc27` `WF4n`

- **Convergence analysis.** This work features a theoretical analysis supported by comprehensive empirical results. `Qg9z` `G6oG`

- **Clean reference to pioneering works.** The references to prior works are easy to follow. `Qg9z`

- **Comprehensive discussion section.** `Qg9z`

- **Possible practical use.** `rcZ7`

---

> ### Author Response · Authors · 2024-12-02
> **Rebuttal Summary (part 2)**
>
> **Concerns and our response.**
>
> - **Misstructured proofs.** The convergence analysis of the proposed method, FeedSign, seems to be built on a wrong assumption. `WF4n`
>
>   - In response, we have checked our paper and found that the relation between the convergence bound and the concerned assumption is vaguely presented. We have revised the analysis part.
>
>   - The reviewer commented, "The revised proof appears convincing".
>
> - **Lack of speed analysis for additional baselines.** Only the ZO methods have a convergence analysis. An analysis of conventional FO methods (like FedAvg) is expected. `G6oG`
>
>   - In response, we added a FedSGD counterpart to our analysis and included discussions and the rationale of the proposed design.
> - **Originality and Significance.** There are existing works on one-bit compression in FL systems. `Qg9z` `rcZ7`
>
>   - In response, we highlighted our differences to the referred works:
>
>     - Our work is built on forward passes-only finetuning; hence, it can be operated with less computation and memory overheads while lowering the communication load to a dimensionally independent 1 bit.
>
>     - In signSGD-like works mentioned by `rcZ7`, 1 bit is elementwise, and the total communication load scales to model size.
>
>     - Apart from the above, the proposed design is resilient against data heterogeneity and Byzantine attacks.
>
>     - We have a different rationale compared to the existing works; existing works address the heavy communication load by separate lossy load compression, while the proposed design directly features a less accurate but much easier commutable gradient estimation.
>
>     - We discussed that the proposed design enjoys some underexplored additional privacy protection.
>
>   - Reviewer `Qg9z` admitted the advantages of our work, but since the discussed features and advantages against the referred earlier works are not perfectly exclusive to the proposed design, concerns remain.
>
>   - We believe that there are observable distinctions between our design and the prior works, and the proposed design can work better under non-ideal settings, as supported by empirical results in the submission and additional results posted in rebuttals.
>
> - **Simple settings regarding Byzantine resilience.** The original setting of 1 Byzantine client in 5 participating clients is not adequate to prove the Byzantine resilience of FeedSign. `G6oG` Besides, some traditional methods, including noise injection and label flipping, are not considered. `rcZ7`
>
>   - In response, we added empirical results with a client pool size of 25 in both language models and vision models.
>
>   - We highlighted that since we adopted a seed-projection pair design, any gradient modification attack boils down to having an inaccurate measure of gradient projection, so the simulated scenario is sufficient.
>
> - **Convergence deficits.** FeedSign converges slower; hence, the communication reduction can not be justified as a significant advantage. `G6oG`
>
>   - In response, we admitted that convergence speed under general settings is not an advantage of FeedSign. Its advantages include better data heterogeneity resilience and Byzantine attack resilience. Additionally, empirical results show that FeedSign converges faster under extreme cases like extreme noniid data distribution or Byzantine attacks.
> - **Practical relevance of Assumption.** Assumption 2, in particular, appears to impose conditions that may not align with realistic settings. `rcZ7`
>
>   - In response, we have added several works empirically verifying the validity of the assumption.
>
> We would like to thank the reviewers, AC, and PC again for their time and efforts.

---

### Meta-Review · Area_Chair_B5dx · 2024-12-20

**Metareview:**

The authors propose a federated fine-tuning approach that just use 1-bit per client to update the model parameters each iteration. The 1-bit update is a carefully-chosen step-size towards a random direction. The size of the step is dependent on the gradient-to-that-random-direction (obtained via function evaluations).

It is a good idea, since such 1-bit updates are pretty good estimators (as well-studied in sketching and 1 bit compressed sensing etc.).

However the novelty beyond this point seems limited as pointed out by all the reviewers. Note that, given signSGD and follow-up papers, robustness/byzantine resilience is not that surprising as the authors claim. In fact, the paper contained a wrong-claim about unbiasedness.

Based on the reviews and discussions, I recommend rejection at this stage.

**Additional Comments On Reviewer Discussion:**

There are fruitful discussion which will potenntially help improve this paper.

---

### Decision · Program_Chairs · 2025-01-22

Reject